# Development of leptospiral virulence-modifying protein detection assay: implications for pathogenesis and diagnostic test development

Reetika Chaurasia,[1,2] Andrea Jacobs,[2] Jie Tang,[2] Songyu Dong,[1] Joseph M. Vinetz[1]

**ABSTRACT**    Leptospirosis, a globally significant neglected tropical disease, continues to lack early and reliable diagnostic methods despite over a century since the discovery of the disease and its etiological agent. Previously, we identified the pathogen-specific paralogous PF07598 gene family encoding virulence-modifying (VM) exotoxins, which play a critical role in leptospirosis pathogenesis. In this study, we developed a monoclonal antibody (mAb)-based capture immunoassay that detects VM proteins in the blood of experimental hamster models, validating the hypothesis that VM proteins function as secretory exotoxins that mediate disease pathogenesis. Monoclonal antibodies were generated against a natural variant, LA0591, a VM protein containing a conserved C-terminal DNase toxin domain but lacking N-terminal ricin B-like lectin domains. Epitope mapping identified specific linear epitopes targeted by mAbs 5F8, 5G10, and 6A5, with distinct binding regions confirmed through binning and peptide mapping. These mAbs demonstrated high-affinity binding to homologous antigens, with sub-picomolar dissociation constants ($K_d$ = 1.41E-09 for 5F8 and $K_d$ <1.0E-12 for 5G10 and 6A5) and cross-reacted with full-length recombinant VM proteins expressed in *E. coli*. Immunoblotting revealed increased expression of VM proteins by *L. interrogans* serovar Copenhageni strain L1-130 under *in vivo*-like conditions. Using mAbs 6A5 and 5F8, a capture ELISA detected circulating VM proteins in the serum and urine from infected hamsters, confirming the secretion of these proteins during infection. This study provides the first evidence of secreted leptospiral exotoxins in the bloodstream of infected animals, advancing the understanding of leptospirosis pathogenesis and establishing a basis for developing novel diagnostic approaches.

**IMPORTANCE** This research addresses the global health issue of leptospirosis, a neglected tropical disease that still lacks early and reliable diagnostic methods despite being known for over a century. The study has developed a novel test using specially designed antibodies to detect specific proteins related to the disease in the blood of infected hamsters. These proteins are linked to the pathogen's ability to cause illness. The successful detection of these proteins in the bloodstream is a significant advancement, as it not only improves our understanding of the disease's progression but also lays the groundwork for developing new diagnostic tools. This could lead to earlier and more accurate diagnoses of leptospirosis, potentially saving lives and reducing the impact of the disease globally.

**KEYWORDS**    monoclonal antibodies, biolayer interferometry, immunoassay, capture ELISA, leptospirosis, *Leptospira*, pathogenesis, diagnosis

P athogenic *Leptospira* are extracellular spirochetes and the causative agents of leptospirosis, a globally significant neglected zoonotic disease (1–3), affecting an

**Peer Reviewer** Amaro Nunes Duarte-Neto, University of São Paulo, São Paulo, Brazil

Address correspondence to Joseph M. Vinetz, joseph.vinetz@yale.edu.

R.C., A.J., J.T., S.D., and J.M.V. declare a potential conflict of interest regarding the publication of this work in so far as they are affiliated with Luna Bioscience, Inc., the principal institutional recipient of the NIH R41 grants that support this work. Some of the work reported here has been filed in patent applications from Yale University. J.M.V and spouse have an equity interest in Luna Bioscience, Inc, which may have a future interest in licensing this work.

See the funding table on p. 18.

estimated 1 million individuals annually, with approximately 60,000 deaths and a case fatality rate of up to 20% in severe cases (4, 5). The clinical presentation ranges from mild illness to severe, life-threatening manifestations. Leptospirosis poses a major public health challenge in tropical and subtropical regions, particularly in areas with inadequate sanitation and during events such as hurricanes, heavy rainfall, and flooding. Its prevalence is anticipated to rise due to the impacts of climate change (6, 7). *L. interrogans*, the species most frequently associated with human infections, is primarily transmitted through contact with urine from infected rodents, particularly rats and mice, where the bacteria persist in the proximal renal tubules (8–10). During the acute illness, the so-called septicemic phase of infection, *Leptospira* remain in the bloodstream (leptospiremia) for at least 10–14 days and then colonize the kidneys (2, 3, 10–13). Agglutinating antibodies can be detected by ELISA as early as 5–7 days following symptom onset; however, this method has limited sensitivity and specificity, often requiring paired pre- and post-infection sera for accurate diagnostic confirmation (14). Standard diagnostic techniques, including the microscopic agglutination test (MAT), culture, and PCR, have inherent limitations, such as false-negative results, particularly in the early stages of infection (15, 16). Antigen-based detection methods offer promise for early diagnosis and broader applicability, especially in resource-limited settings (17, 18).

We previously identified the paralogous PF07598 gene family encoding VM proteins, whose transcripts are upregulated both *in vitro* under host-mimicking conditions and *in vivo* in small animal models of acute infection (11, 19), supporting the hypothesis of their critical role in leptospirosis pathogenesis. VM proteins are secreted exotoxins, characterized by secretory signal peptides, two N-terminal tandemly repeated R-type lectin domains (RBLs) (except for the LA0591 ortholog, which lacks RBLs), and a C-terminal toxin domain with DNase activity (20, 21). DNase activity has been experimentally demonstrated for a limited subset of recombinant VM proteins (LA3490, LA0591, LA0620, LA1400, and LA1402). However, the molecular mechanisms driving this activity, as well as the specific amino acid residues responsible, remain to be elucidated. Although putative critical residues have been suggested in earlier studies, their involvement in DNase activity has yet to be experimentally confirmed.

Vaccination studies in mice demonstrated that VM proteins significantly reduced bacterial loads ($10^4$-fold to $10^5$-fold lower) in key organs such as the liver and kidneys and prevented mortality, highlighting their therapeutic potential (22). Strain variations within *Leptospira* species influence infection dynamics and host interactions. For example, Putz et al. demonstrated distinct disease profiles in hamsters infected with *L. borgpetersenii* serovar Hardjo strains JB197 and HB203, which share 99% genome similarity: JB197 induces severe acute infection, whereas HB203 causes asymptomatic chronic infection (23). JB197 showed significant upregulation of the Q04V07 VM protein (LIC12399 ortholog) at 29°C and 37°C compared with HB203 (23).

Recent use of CRISPR-dCas9 knockdown of the LIMLP11655 (VMP) in *L. interrogans* serovar Manilae supports the hypothesis (and our previously published work) that VMPs are a key virulence factor in the cellular pathogenesis of leptospirosis (24), which underscores the potential critical role of the PF07598 gene family in leptospiral pathogenesis (11, 19–21). The same group recently used dual RNA-Seq to study how *L. interrogans* affects host and pathogen gene expression during infection. They found that only two of 12 *L. interrogans* serovar Manilae PF07598-encoded VM proteins were expressed at the transcriptional level (*L. interrogans* serovar Manilae nomenclature, *LIMLP_11655* and *LIMLP_11660;* orthologs in *L. interrogans* serovar Lai, *LA1400 and LA1402,* and serovar Copenhageni *LIC12340 and LIC12339,* respectively). *LIMLP_11660* inactivation led to complete loss of virulence, which complementation restored (25). These two VM protein homologs were associated with epithelial cell tight junction disruption, increased calcium balance (25). These independent studies validated our hypothesis and previous findings, underscoring that VMPs are secreted protein exotoxins.

The goal of this study was to develop a capture ELISA-based immunoassay to detect pathogen-specific VMP antigens in experimental animal samples to advance early diagnostics for leptospirosis. Detecting VMPs in the blood, tissues, and urine of infected animals would support the hypothesis that these leptospiral-secreted exotoxins are circulating and would contribute to the systemic clinical manifestations of this infection, including "vasculitis"-like syndromes, shock, and pulmonary hemorrhage (2, 3, 10, 26). Our findings demonstrate the presence of leptospiral VM proteins in the serum and urine of infected hamsters, supporting the hypothesis that circulating VM proteins mediate disease pathogenesis.

## MATERIALS AND METHODS

### Leptospira growth and maintenance of low-passage strains in hamsters

*Leptospira* was maintained at 30°C in semisolid Ellinghausen, McCullough, Johnson, and Harris medium (EMJH, BD Biosciences, USA) and cultured in liquid EMJH medium (27). Growth was regularly monitored using a darkfield microscope (Nikon Eclipse E600, Japan). Mid-logarithmic cultures in liquid EMJH medium were harvested by centrifugation at $12,000 \times g$. The pelleted cells were washed twice with 1× PBS (pH 7.4). Outbred Syrian Golden hamsters (Jackson Laboratories, ME, USA) were inoculated with $2.5 \times 10^8$ cells suspended in 1 mL of 1× PBS.

Hamster experiments were conducted under Animal Biosafety Level (ABSL-2) conditions with approval from Yale University's Institutional Animal Care and Use Committee (Protocol 2022-20243). Three-week-old female *Leptospira*-negative hamsters were obtained from Jackson Laboratories (ME, USA) and housed in a specific-pathogen-free environment. They were kept in individually ventilated cages with sterile bedding changed twice weekly and provided with food and water *ad libitum* throughout the study. All procedures minimized pain and distress under veterinary supervision.

Groups of five 4- to 6-week-old male hamsters were injected intraperitoneally (IP) with $2.5 \times 10^8$ leptospires/mL of *L. interrogans* serovar Copenhageni strains L1-130 in 1 mL of 1× PBS. The hamsters were monitored twice daily for signs of illness such as appetite loss, lethargy, breathing difficulties, prostration, ruffled fur, and 10% wt loss. Those showing severe symptoms were euthanized using $CO_2$, per Association for Assessment and Accreditation of Laboratory Animal Care (AAALAC)/American Veterinary Medical Association (AVMA) guidelines, and classified as having severe or lethal leptospirosis. Infected hamster blood, lungs, liver, and kidneys were collected aseptically and maintained at 30°C in semisolid EMJH containing 5 fluorouracil (5FU, 200 µg/mL) and neomycin (4 µg/mL). Retroorbital blood and urine samples were collected and processed at the scheduled time point and stored at −80°C until the analysis of VM proteins. Whole blood was allowed to clot by incubating the tubes for 20 min at room temperature, and serum was separated by centrifugation at $1,500 \times g$ for 10 min at 4°C. Urine samples were collected from hamsters at the time of euthanasia by directly aspirating urine from the bladder through an open abdominal cavity to avoid contamination. Samples were then centrifuged at $28,000 \times g$ for 30 min to separate cellular debris and particulate material. Both the supernatant and pellet fractions were stored for downstream analysis. For the detection of VM, we primarily analyzed the pellet fraction, which was resuspended in 100 µL of 1× PBS, as preliminary experiments indicated that VM was predominantly associated with particulate components, possibly due to secretion in vesicles or association with host/bacterial cells. Both the blood and urine were aliquoted to avoid freeze-thaw cycles and stored at −80°C until further use.

### Cloning, recombinant protein expression, and purification

Recombinant proteins were expressed and purified as published (21, 22, 28). Briefly, the constructs encoding the PF07598 proteins—LA3490 (Uniprot: Q8F0K3), the RBL1 and RBL1+2 (RBLs) domains of LA3490, and LA1402 (Q8F6A7) from *L. interrogans* serovar Lai

—as well as LIC12340 (Q72*P* X 7), the Lai ortholog of LA1400, and LIC12985 (Q72N53), a natural variant of the Lai ortholog LA0591 that encodes only the C-terminal region of the VM exotoxin, from serovar Copenhageni L1-130, were obtained from laboratory stocks (21, 22, 28). To generate the LA0591 mutant, the following site-directed mutations were introduced into the protein sequence: at position 203, arginine (R) was replaced with lysine (K); at position 205, histidine (H) was substituted with alanine (A); at position 221, threonine (T) was replaced with alanine (A); at position 222, arginine (R) was replaced with lysine (K); and at position 254, arginine (R) was substituted with lysine (K). These amino acid substitutions were designed to assess the potential structural and functional effects of altering charge and polarity at these specific residues. Post-signal coding sequences were synthesized and cloned into pET32b(+) (Gene Universal Inc., USA). Constructs for LA3490, LA1402, RBL1, and RBLs were fused to mCherry (AST15061.1) via a (Gly$_4$Ser)$_3$ linker flanked by enterokinase sites, whereas full-length LA1400 and LA0591 were cloned without mCherry; all constructs were sequence- and orientation-verified by restriction digestion and sequencing (21, 22, 28).

These VM proteins are cysteine-rich, and the recombinant proteins were expressed in SHuffleT7 competent *E. coli* cells (New England Biolabs, USA) owing to their capacity to promote disulfide bonds in the cytoplasm, ensuring proper protein folding. Transformants were sub-cultured into Luria-Bertani (LB) medium containing 100 µg/mL ampicillin at 37°C . When cultures had reached an OD of 0.6, expression was induced at 16°C and 250 rpm for 24 h via the addition of 1 mM isopropyl-*b*-D-thiogalactoside (IPTG; Sigma-Aldrich, USA). Following induction, the cells were pelleted by centrifugation and then lysed in CelLytic B (Cell Lysis Reagent; Sigma-Aldrich, USA) containing 50 units/mL benzonase nuclease (Sigma-Aldrich, USA), 0.2 mg/mL lysozyme, and EDTA-free protease inhibitor cocktail (Roche, USA) plus 100 mM PMSF (Sigma-Aldrich, USA) for 30 min at 37°C. Lysates were centrifuged at 4°C and 18,514 × *g* for 10 min. Supernatants and pellets were separated and then analyzed by 4%–12% bis-tris sodium dodecyl sulfate-polyacrylamide gel electrophoresis (SDS-PAGE). Protein concentrations were determined by BCA assay (Pierce BCA Protein Assay Kit, Thermo Fisher Scientific, USA). These recombinant proteins were purified using a 1 mL pre-packed Ni-Sepharose AKTA HisTRAP column (GE Healthcare, USA) pre-equilibrated with a buffer containing 100 mM NaH$_2$PO$_4$, 10 mM Tris-HCl, 25 mM imidazole, pH 8.0. Bound fusion protein was then eluted from the column in the presence of 500 mM imidazole, pH 8.0. Eluates were pooled and concentrated via a 30 kDa Amicon Ultra centrifugal filter, and recombinant protein preparations were dialyzed overnight against 1xPBS (pH 7.4) with gentle stirring (350 rpm) at 4°C (10 kDa cutoff, Slide-A-Lyzer, Thermo Scientific, USA), followed by size exclusion via a 7 kDa Zeba desalting spin column (Thermo Fisher Scientific, USA) to remove imidazole and then stored at −80°C until use.

## Sodium dodecyl sulfate-polyacrylamide gel electrophoresis (SDS-PAGE) and western immunoblot analysis

SDS-PAGE was performed according to Laemmli's method (29). Proteins were transferred to nitrocellulose membranes, which were then blocked for 2 h with 5% nonfat dry milk dissolved in 1× PBST buffer (AmericanBio, USA), and then probed with mouse monoclonal antibodies (1:1,000 dilution). After washing three times with PBST, membranes were incubated for 2.5 h with alkaline phosphatase-conjugated goat anti-mouse IgG (H + L) as the secondary antibody (KPL, USA) at a dilution of 1:5,000 in PBST. Blots were developed in 5-bromo-4-chloro-3-indolyl phosphate and nitroblue tetrazolium solution (BCIP/NBT; KPL, USA).

## Generation of monoclonal antibodies

Monoclonal antibodies targeting the post-signal full-length LA0591 were generated in mice by Precision Antibody, Columbia, Maryland, USA.

## Bio-layer interferometry

Bio-layer interferometry (BLI), a label-free optical analytical technology (commercially known as Octet), was used for measuring antigen and antibody interactions and kinetics. Mouse monoclonal antibody (mAb) was captured using anti-mouse $F_c$ capture (AMC) dip-and-read biosensors. These probes were then dipped into wells containing the LA0591 antigen at a concentration of 500 nM to measure antigen-antibody association rate, $K_d$. Binding affinity ($K_d$) is expressed as the equilibrium dissociation constant (unit: M).

Following this, the probes were placed in a PBS assay buffer to determine the dissociation rate (off rate). The dissociation constant ($K_d$) was calculated using the association rate constant ($K_a$, units: 1 /M·s) and the dissociation rate constant ($K_d$, units: $s^{-1}$) through 1:1 local fit analysis in Fortebio data acquisition software v. 8.0.0.99.

The equilibrium dissociation constant ($K_d$, units: M) quantifies the binding affinity between the antibody and antigen. The dissociation rate constant ($K_d$ units: $s^{-1}$) describes the rate at which the antibody-antigen complex dissociates, whereas the association rate constant ($K_a$ units: 1 /M·s) represents the rate at which the antibody binds to the antigen. The $R^2$ value indicates the goodness-of-fit between the experimental data and the fitted curve, whereas the $\chi^2$ value reflects the degree of error between them. A $\chi^2$ value between 1 and 2 is considered accurate, with values below 1 indicating high accuracy (Table 1).

## Binning matrix/pairing analysis

This was performed using a label-free BLI-based sandwich assay. Clones were captured from the culture supernatants to a 1.0 nm threshold using immobilized goat anti-mouse Fc antibody on the biosensor. Quenching was performed with a F(ab')2 fragment of anti-mouse IgG (50 µL/well in PBS) to block unoccupied binding sites on the sensor. Capture/binding was performed by adding the target LA0591 at 500 nM in solution. Self-pairing was performed with the capture antibody as a baseline. Binding of a second mAb to the bound target LA0591 was detected, and multiple pairing sets were arranged. Dilutions were made in PBS. Abs that pair are highlighted in green. Self-pairing or blocking effect is highlighted in red and is used as the threshold to determine the strong pairs. Mouse-pooled purified IgG was included as a negative control as a detection antibody, as well as to show that the sensor is completely saturated.

## Linear epitope mapping using overlapping synthetic peptides

A custom-made Mimotopes peptide library (Pepsets) for post-signal LA0591 (291 aa) was commercially synthesized by Mimotopes, USA, on their unique proprietary SynPhase Lanterns-based Multipin system on a 96-well plate. The library contained 71 peptides in the format of Biotin-Spacer-Peptide-COOH. Each peptide is 12 aa plus an SGSG spacer,

**TABLE 1** Binding kinetics of monoclonal antibodies against target antigen, recombinant leptospiral virulence-modifying protein, LA0591[a,b,c,d,e]

| Clones | Ab loading response in nm shift | Con. (nM) | Ag binding response in nm shift | KD (M) | $k_{on}$ (1 /Ms) | $k_{dis}$ (1 /s) | Full $X^2$ | Full $R^2$ |
|---|---|---|---|---|---|---|---|---|
| 6A5 | 0.2086 | 500 | 0.1109 | <1.0E-12 | 2.05E+04 | <1.0E-07 | 0.0085 | 0.9742 |
| 5G10 | 1.1788 | 500 | 0.2255 | <1.0E-12 | 7.54E+03 | <1.0E-07 | 0.0324 | 0.9884 |
| 5F8 | 1.2214 | 500 | 0.3633 | 1.41E-09 | 9.84E+03 | 1.38E-05 | 0.0059 | 0.999 |
| 5E10 | 0.891 | 500 | 0.3176 | 4.19E-09 | 1.26E+04 | 5.27E-05 | 0.0076 | 0.998 |
| 5A7 | 1.1815 | 500 | 0.1297 | 5.92E-08 | 5.17E+03 | 3.06E-04 | 0.0036 | 0.995 |

[a]KD (M): Equilibrium dissociation constant.
[b]$k_{on}$, on-rate constant, and $k_{dis}$, dissociation-rate constant, which have units of $M^{-1} s^{-1}$ and $s^{-1}$, respectively.
[c]This statistic quantifies the overall difference between a model's predictions and the observed experimental data.
[d]A smaller "Full $X^2$" value indicates a closer alignment of the model with the data.
[e]A higher "Full $R^2$" value signifies that the model explains a greater proportion of the observed variability.

supplied as lyophilized, and stored at −20°C. The plate was blocked with 5% bovine serum albumin (BSA, Sigma-Aldrich, USA) in PBST for 2 h. Monoclonal or vaccinated hamster sera (against ΔLA0591 +LA1402) were added to the well at a 1:2,000 dilution in 100 µL of 1× PBS and incubated for 1 h at 37°C. Wells with PBS served as a negative control. The wells were washed four times with PBST to remove the non-specific binding and then incubated with either anti-mouse-IgG (5F8 isotype-IgG1, 6A5 isotype-IgG2β)-HRP conjugate or anti-hamster-IgG-HRP at a 1:10,000 dilution in 1× PBS (KPL, USA) for 1 h at 37°C. The reaction was developed with 100 µL of ready-to-use TMB substrate (Sigma-Aldrich, USA) and stopped with 2 M $H_2SO_4$. The absorbance (optical density, OD) was read at 450 nm using a SpectraMax M2e Microplate Reader (Molecular Devices, USA).

## Epitope mapping using PyMOL visualization

Three-dimensional structures of LA0591 (Uniprot ID: Q8F8G6, Predicted AlphaFold: AF-Q8F8G6-F1) and full-length LA3490 (Uniprot ID: Q8F0K3, Predicted AlphaFold: AF-Q8F0K3-F1) were analyzed using PyMOL version (TM) 3.1.3.1 to localize conserved immunoreactive epitopes. Epitope 19 (NSHGPLQGGGYF), Epitope 20 (PLQGGGYFFNTA), and Epitope 67 (NRRGSGGYPTSA) were color-coded in red, blue, and green, respectively, and mapped onto the surface structures of each protein.

## Multiple sequence alignment

Peptide epitopes of mAbs and paralogs PF07598 gene family encoding VM exotoxins from *L. interrogans* serovars Lai (*n* = 12) and Copenhageni L1-130 (*n* = 13) and their orthologs in *L. borgpetersenii* (two paralogs, *1:* orthologs of LIC12844; three identical copies, and *2:* LIC12399) were sequentially aligned using pairwise MUSCLE protein alignment using default settings in Jalview V2.11.4. (19151095). Percentage identity was determined using pairwise alignment in Jalview V2.11.4.

## Leptospiral expression of virulence-modifying proteins determined by western immunoblot

Previously, we demonstrated that VM proteins are transcriptionally upregulated *in vivo* in a hamster model of acute leptospirosis (19). To optimize their expression *in vitro*, *Leptospira* was grown under conditions mimicking the *in vivo* host environment, which is known to induce virulence gene expression (30). Mid-logarithmic cultures in unmodified EMJH medium were harvested by centrifugation at 12,000 × *g*. Pelleted cells were washed twice with 1× PBS, resuspended in liquid EMJH medium supplemented with 120 mM NaCl (Sigma Aldrich, USA), and then incubated at 37°C for 4 h. The pelleted cells were resuspended in 200 µL of Bug-buster reagent (Sigma Aldrich, USA) containing protease inhibitor cocktail (Merck Millipore, Germany), and the cell-free lysate was separated by centrifugation at 12,000 × *g* for 10 min at 4°C. Induced and uninduced (corresponding to baseline *in vitro* expression) cell-free lysates were analyzed by western blot probed with monoclonal antibodies as above. Total protein was estimated by BCA assay (Pierce BCA Protein Assay Kit, Thermo Fisher Scientific, USA).

## Indirect ELISA

Indirect ELISA was performed as previously described (31) using 96-well ELISA plates (Corning, USA). Plates were coated with 100 ng of VM antigens in a molar ratio (LA0591 and RBLs) per well in 100 µL of 0.05 M sodium carbonate buffer, pH 9.6, and incubated overnight at 4°C. The plate was blocked with 5% bovine serum albumin (BSA, Sigma-Aldrich, USA) in PBST for 2 h, followed by incubation with 100 µL of mAbs in 2-fold serial dilutions starting at 1 µg/mL (5F8, 6A5, or 5G10 generated against LA0591) in PBS with 5% BSA for 1 h, at 37°C. The wells were washed four times with PBST to remove the non-specific binding and then incubated with anti-mouse-IgG-HRP conjugate (1:10,000; KPL, USA) for 1 h, at 37°C. The reaction was developed with 100 µL of ready-to-use TMB substrate (Sigma-Aldrich, USA) and stopped with 2 M $H_2SO_4$. OD was read at 450 nm

using a SpectraMax M2e Microplate Reader (Molecular Devices, USA). Recombinant LA0591 served as a positive control, and RBLs and secondary antibodies served as a negative control.

## Optimization of capture ELISA

A capture ELISA was optimized for evaluating antibody pairs and the specific detection of VM antigens in serum, blood, and urine by performing serial dilutions of coating mAbs at 1 μg/mL and a constant 100 ng/well concentration of control recombinant proteins (LA0591 and RBLs) with the following combinations: (i). 6A5 as the capturing antibody and 5F8 as the detecting antibody, (ii) 5F8 as the capturing antibody and 6A5 as the detecting antibody, and (iii) 5G10 as the capturing antibody and 6A5 as the detecting antibody. Further optimization was carried out by varying the antigen concentrations and testing 1 μg/mL of different pairs of capturing and detecting antibodies: (i) 6A5 as the capturing antibody and 5F8 as the detecting antibody, and (ii) 5F8 as the capturing antibody and 6A5 as the detecting antibody. A diluted capture antibody was added (100 μL/well) to the 96-well ELISA plate (Corning, USA) and incubated at 4∘C overnight. The plate was washed four times with 1× PBST and blocked (300 μL/well) with 5% BSA (Sigma-Aldrich, USA) in PBS and incubated at 37∘C for 2 h. The plate was washed and incubated with VM antigens for 1 h at 37°C. After washing the plate four times with 1xPBST, the detecting monoclonal antibody (1:5,000) was added to the wells, and the plate was incubated for 1 h at 37°C. The plate was incubated with HRP-conjugated goat anti-Mouse IgG (H+L) (Sigma, Missouri, USA) diluted 1:10,000 in 0.05 % PBST, added to the ELISA plate (100 μL/well), and incubated at RT for 1 h. After four washes, 100 μL of TMB (Sigma, USA) was added, and the reaction was stopped with 2M $H_2SO_4$.

## Biotinylation of detecting 5F8 monoclonal antibody

Purified 5F8 mAb was biotinylated using a Pierce Antibody Biotinylation Kit (Thermo Scientific) according to the manufacturer's instructions. Briefly, a stock of 8.5 mM biotin was prepared by adding 100 μL of ultrapure water to a 1 mg EZ-Link NHS-PEG4-biotin containing microtube. The concentration of 5F8 mAb (MW 150 kDa) was adjusted to 1 mg/mL. To achieve a 50-fold excess of biotin, 7.84 μL of the Biotin stock was added to 200 μL of 5F8 mAb and mixed by gently pipetting up and down. The tube was incubated for reaction at room temperature for 30 min and then desalted with a 7 kDa Zeba Spin Desalting Column (Thermo Scientific), and antibody biotinylation was confirmed by the Antibody Biotinylation Check Kit (Abcam, USA)

## Detection of VM exotoxins in experimentally infected hamster blood samples

The optimized pair of 6A5 capturing and 5F8 detecting antibodies was used to screen and detect VM exotoxins in hamster serum samples. Briefly, 1 μg/mL of capture antibody was added (100 μL/well) to the 96-well ELISA plate (Corning, USA) and incubated at 4∘C overnight.

The plate was blocked with 5% BSA (Sigma-Aldrich, USA) in PBS and incubated at 37∘C for 2 h. The plate was washed four times with PBST and incubated with hamster serum/urine at a 1:50 dilution for 1 h at 37°C. After four washes with 1× PBST, the plate was incubated with 100 μL/well of 50 mol of biotinylated detecting 5F8 monoclonal antibody (1 μg/mL) for 1 h at 37°C. The plate was incubated with streptavidin-HRP (1:200, R&D SYSTEMS, USA) for 1 h at 37°C. The reaction was developed with 100 μL of 3,3′,5,5′-tetramethylbenzidine (TMB) (Sigma, USA) and stopped with 2M $H_2SO_4$. Recombinant LA0591 served as a positive control, and assay diluent served as a negative control. The mean (x) and standard deviation (SD) of $OD_{450}$ nm of all samples were calculated. The data were interpolated using a measured quantity of LA0591 in a standard curve.

## Statistical analysis

Experiments were performed in triplicate and repeated twice. Data are presented as mean ± SD and were analyzed by the non-parametric Mann–Whitney test to determine significant differences between individual groups and were considered statistically significant when $P < 0.05$. Analyses and graphs were generated using Graph Prism version 8 (GraphPad Software, Inc., USA). Final figures were generated in Illustrator version 25.2.

## RESULTS

### Quantitative analysis of the binding kinetics of newly developed anti-virulence-modifying protein monoclonal antibodies

VM exotoxins are encoded by single genes as polypeptides that are potentially oligomerized into fully active exotoxins (~640 amino acids) (20, 21). Oligomerization is a well-documented mechanism of activation for various bacterial exotoxins, including anthrax toxin (32) and listeriolysin O (33). However, direct evidence for homo-oligomer formation by VM exotoxins remains to be established and warrants further investigation.

The PF07598 gene family typically encodes variants possessing both carbohydrate-binding receptor (RBL) domains and a C-terminal toxin domain. In contrast, LA0591 is a unique natural variant that lacks the RBL domains entirely and encodes only the C-terminal toxin domain, distinguishing it from other family members and suggesting potentially distinct functional roles (Fig. 1). Due to the advantage of encoding only the C-terminal domain, LA0591 was chosen for monoclonal antibody generation. A scouting experiment was conducted using five clones against the 500 nM concentration of target antigen LA0591. All five clones tested positive, with binding affinities ranging from pM to nM. The affinity ranking ($K_D$) from the highest to the lowest was 6A5 > 5G10 >5F8 >5E10 >5A7 (Table 1). Epitope binning revealed that clone 6A5 recognizes a unique epitope and can pair with 5F8, 5E10, and 5G10 in both capture and detection antibody formats. Clones 5G10 and 5F8 may share the same or overlapping epitope but can pair with 5E10 and 6A5 in both formats. Notably, 5E10 also appears to target a unique epitope and can pair with 5F8, 5G10, and 6A5 in capture and detection formats. Clone 5A7 exhibited the lowest affinity toward the target antigen (Table 2). Clone 5E10 did not yield sufficient mAb for scale-up. Therefore, we selected clones 5G10, 5F8, and 6A5 for large-scale production, as they demonstrated good yield, strong binding affinity, and consistent reproducibility.

### Determination of an optimal concentration for ELISA

Both indirect and capture ELISA were optimized by assessing and confirming the dilution of antigens and mAb pairs. In the indirect ELISA, 2-fold serial dilutions of 5F8, 6A5, and 5G10 mAbs were tested against a fixed amount of antigen, resulting in detection limits of

**TABLE 2** Biolayer interferometry (Octet) analysis of monoclonal antibody pairing using as binding antigen recombinant leptospiral virulence-modifying protein, LA0591[a,b,c,d]

| Capture mAb in pair | Detection mAb in pair | | | | |
|---|---|---|---|---|---|
| | 6A5 | 5E10 | 5G10 | 5F8 | 5A7 |
| 6A5 | 0.04 | 0.3246 | 0.2652 | 0.3658 | 0.0399 |
| 5E10 | 0.1099 | 0.0304 | 0.1584 | 0.0575 | 0.1675 |
| 5G10 | 0.0835 | 0.1359 | 0.0518 | 0.1365 | 0.0912 |
| 5F8 | 0.0783 | 0.0075 | 0.1019 | 0.0283 | 0.128 |
| 5A7 | 0.0929 | 0.2232 | 0.1762 | 0.2364 | −0.0076 |

[a]Binding Response Unit (nm) – in nanometers, measures wavelength shift due to biomolecular binding (reflecting mass changes on the biosensor surface).
[b]Paired antibodies are indicated by an underline.
[c]The self-pairing or blocking effect, highlighted in gray, is used as a threshold to determine the strong pairs.
[d]Mouse-pooled purified IgG was included as a negative control as a detection antibody, and it shows that the sensor is completely saturated.

**(A)**

12/13 VMPs in Lai/Copenhageni
(~640 aa)

**(B)**

LA0591/LIC12985 (313 aa),
Naturally lack RBLs (RBL1+RBL2)

**(C)**

Superimposition of CTD
of LA3490 and LA0591

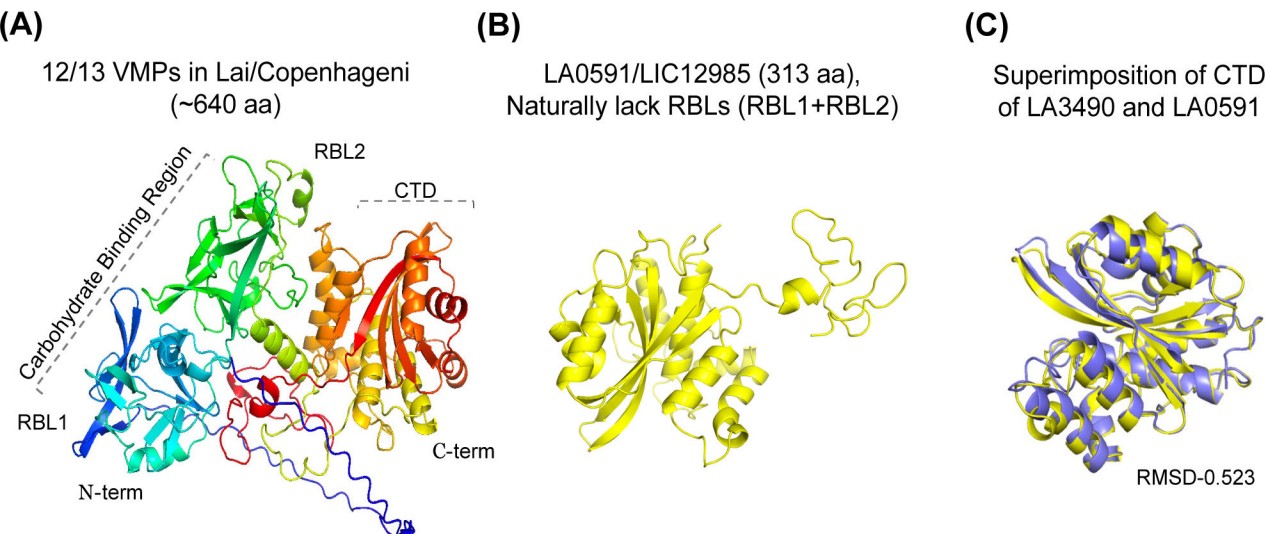

RMSD-0.523

**FIG 1** Three-dimensional structure organization of paralogous PF07598 gene family VM proteins. (A) *L. interrogans* encoding LA3490 shows the representation of the AlphaFold 3D-generated model of full-length VM proteins (~640 aa), comprising a multi-globular domain N-terminal to C-terminal (blue to red color) residues visualized in PyMOL 2.4.0 https://pymol.org/2/. VM proteins are predicted with high confidence to have two tandemly repeated, N-terminal ricin B-like (RBL) lectin domains. N-terminal (RBL1) *β*-trefoil folds identified as ricin B domains. (B) *L. interrogans* naturally encodes a variant VM protein, LA0591, which lacks the N-terminal domains (RBL1 and RBL2) and consists only of a C-terminal domain (313 amino acids). (C) Structural alignment between the full-length LA0591 protein and the C-terminal region of LA3490 suggests an RMSD of 0.523.

1.98 ng/mL for 5F8, 8.56 ng/mL for 6A5, and 0.33 µg/mL for 5G10 (Fig. 2A). In the capture ELISA, 2-fold serial dilutions of capture (starting at 1 µg/mL) and detection antibodies (diluted 1:2,000) were used to detect LA0591, with different mAb pairs tested: capture 6A5/detection 5F8 (5.2 ng/mL), 5G10/6A5 (14.5 ng/mL), and 5F8/6A5 (10.2 ng/mL) (Fig. 2B). The two selected pairs, 6A5/5F8 and 5F8/6A5, were further evaluated using 2-fold serial dilutions of LA0591 (starting at 1,000 ng/mL), with detection limits of 1.84 ng/mL for 6A5/5F8 and 1.31 ng/mL for 5F8/6A5 (Fig. 2C and D). Subsequent experiments utilized 6A5/5F8 (capture/detection) because 6A5 exhibited the highest binding affinity, as confirmed by both indirect and capture ELISA. RBLs with mAb mixtures were used as a negative control. Although the overall binding signal for the 6A5/5F8 pair appeared higher in Fig. 2C compared with 5F8/6A5 in Fig. 2D, the calculated LoD was based on the lowest antigen concentration that consistently produced a signal exceeding the background (mean of blank +3 SD). The limit of detection (LoD) in the indirect ELISA was estimated based on the lowest concentration of antibody that produced a signal significantly above the background (mean of blank wells + 3 SD). Although a fixed antigen concentration was used, serial dilutions of the antibodies allowed us to assess the sensitivity of each mAb in detecting the immobilized antigen. This approach reflects the minimum concentration of antibody required to generate a detectable signal under the given assay conditions. Minor variations in background noise and signal consistency across replicates may have influenced the calculated LoD values. Therefore, although 6A5/5F8 showed stronger signal intensity, the 5F8/6A5 pair demonstrated slightly better sensitivity under our assay conditions.

## Cross-reactivity of mAbs among recombinant and native VM proteins

Exposure to 120 mM NaCl in EMJH medium was employed to simulate the physiological osmolarity conditions encountered by *Leptospira* during host infection. The transition from low-salt environmental conditions to the higher osmolarity of mammalian tissues is known to influence the expression of virulence-associated genes and host-adaptive responses. Supplementation of EMJH medium with NaCl thus provides a relevant *in vitro* model for studying *Leptospira* under host-mimicking conditions (30).

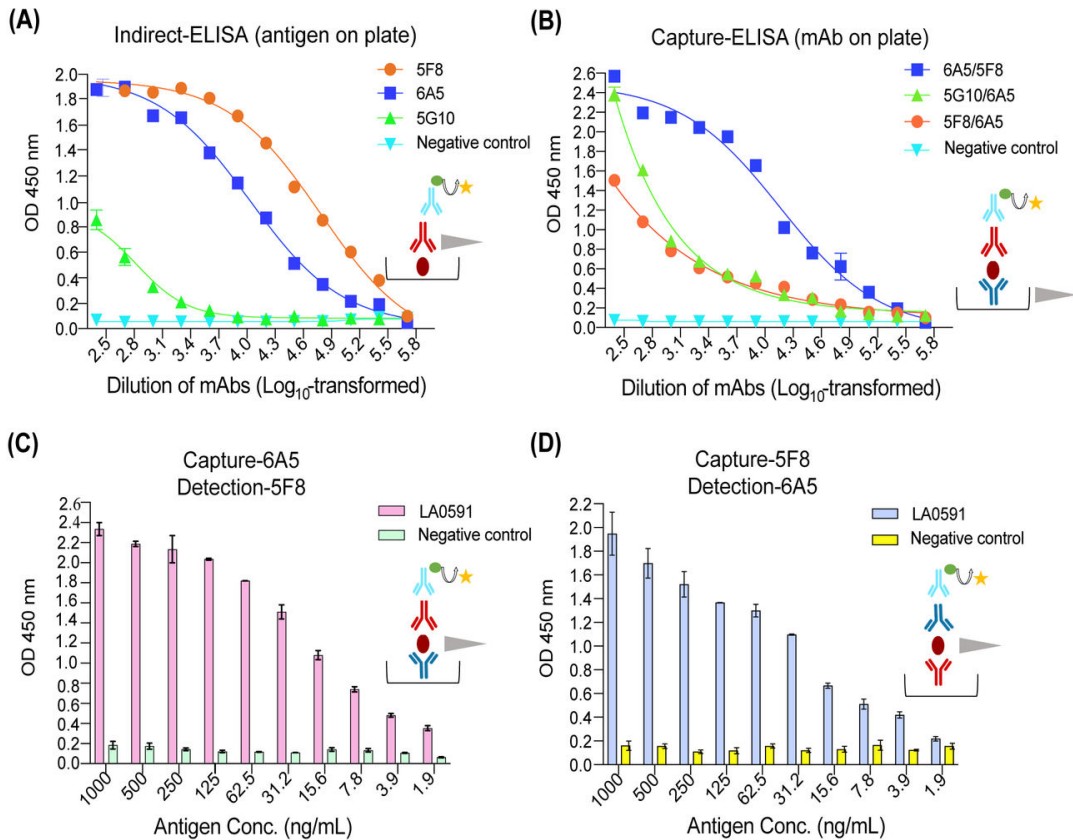

**FIG 2** Optimization of indirect and capture ELISA. (A) The LA0591 protein, at an initial concentration of 1,000 ng/mL, was 2-fold serially diluted and coated onto a 96-well ELISA plate (Corning, USA), followed by overnight incubation at 4°C. Monoclonal antibodies 5F8, 6A5, and 5G10 were added at a 1:2,000 dilution in PBST. Goat anti-mouse IgG-HRP conjugate (mAbs 5F8 and 5G10: IgG1γ isotype, mAbs 6A5: IgG2β isotype) was used at a 1:10,000 dilution. (B) Capture ELISA was optimized with various pairs of capture/detection antibodies (6A5/5F8, 5G10/6A5, and 5F8/6A5). The capture antibody was coated on a 96-well plate at a concentration of 1 µg/mL, followed by a 2-fold serial dilution. The antigen was used at 1,000 ng/well, with the detection antibody added at a 1:2,000 dilution in 1 × PBS. (C) The capture/detection antibody pair was 6A5/5F8. (D) The capture/detection antibody pair was 5F8/6A5. (C and D) Capture ELISA was optimized with antigen concentrations ranging from 1,000 to 1.9 ng/well in 2-fold serial dilutions. Both capture and detection antibodies were diluted 1:2,000 in 1× PBS. RBLs incubated with a mixture of 5F8, 6A5, and 5G10 served as negative controls. The gray arrow represents the serially diluted variable concentration of either antigens or antibodies.

Western blot analysis confirmed the binding affinity, specificity, and cross-reactivity of the mAbs with recombinant and native VMPs. mAb 5A7 was specific to LA0591, whereas 5F8, 5G10, and 5E10 cross-reacted with the full-length LA3490 but not with LA1402 or LA1400, which are ancestral VM proteins in the evolution of the PF07598 gene family. mAb 6A5, with the highest binding affinity ($K_D$ <1.0E-12), showed broad reactivity with all tested recombinant proteins, including LA3490, LA1402, LA1400, and LA0591 (Fig. 3A). The observation that 6A5 recognizes a single band of LA0591, whereas other mAbs detect multiple bands, likely reflects differences in epitope recognition and binding specificity (Fig. 3A; Fig. S1). RBL1 was used as a negative control (Fig. 3A). These findings demonstrate that the mAbs bind to recombinant VM proteins and recognize distinct epitope sites.

These mAbs also recognize native VM paralogous proteins (Fig. 3B) of pathogenic *L. interrogans* serovars Lai, Copenhageni-L1-130, Canicola, and human isolate SL-23 (34) were assessed for the expression of VMPs. Non-pathogenic *L. biflexa* serovar Patoc does not encode for VM paralogous proteins and therefore serves as a negative control. Although the epitopes recognized by mAbs 5F8/5G10 and 6A5 are conserved in the Lai strain, the lack of reactivity observed may be due to low expression levels of the VMPs under the tested conditions, even after NaCl treatment (Fig. 3B). It is also possible

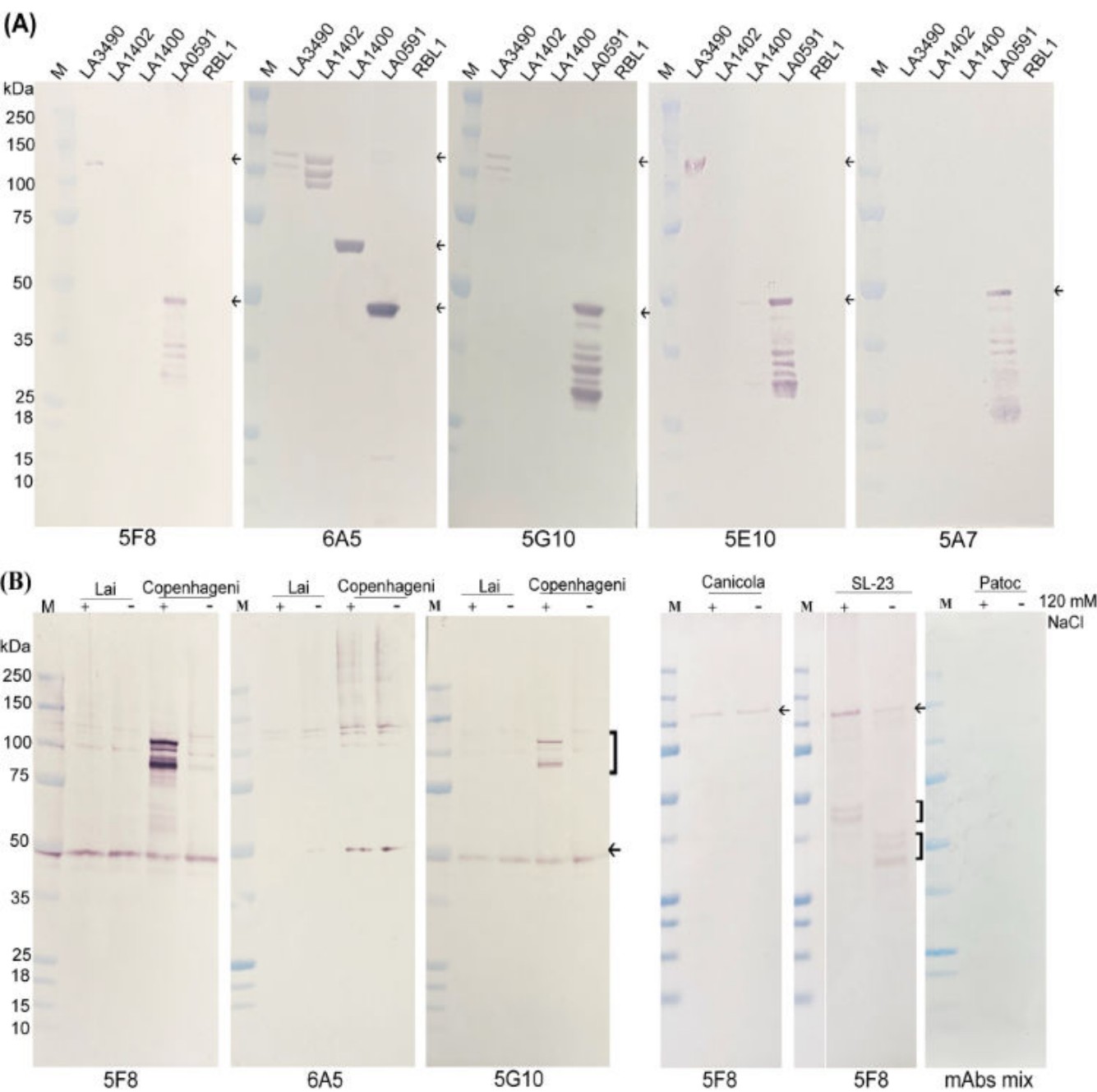

**FIG 3** Reactivity of murine monoclonal antibodies with paralogous VM proteins. (A) Soluble recombinant VM proteins purified using AKTA—including LA3490 (123 kDa; residues 19-639 aa, with mCherry), LA1402 (123 kDa; residues 28-641 aa, with mCherry), LA1400 (70 kDa; residues 1-573 aa, without mCherry), LA0591 (48 kDa; residues 23-313 aa, without mCherry), and RBL1 (52 kDa; residues 40-147 aa, with mCherry)—were analyzed by 4%–12% SDS-PAGE (21, 22, 28). Following electrophoresis, proteins were transferred to a nitrocellulose membrane and probed with mouse monoclonal antibodies 5F8, 6A5, 5G10, 5E10, and 5A7 (1:1,000 dilution in 1× PBST), incubated overnight at 4°C. Detection was performed using goat anti-mouse IgG-ALP conjugate (KLP, USA) at a 1:5,000 dilution. M indicates the molecular weight marker. Arrows indicate the positions of the respective protein bands. (B) *Leptospira* was grown in EMJH liquid media uninduced (−) and induced (+) with 120 mM sodium chloride to achieve physiological osmolarity and induce the expression of virulence genes (22). Fifteen micrograms of uninduced (−) and induced (+) cell-free lysates from pathogenic serovars Lai, Copenhageni, Canicola, SL-23, and non-pathogenic serovar Patoc were separated by 4–12% SDS-PAGE, followed by immunoblotting with mAbs 5F8, 5G10, and 6A5. The upregulation of VM proteins in the serovar Copenhageni strain Fiocruz L1-130 is indicated by parentheses, whereas the arrows represent cross-reactive VMPs. Patoc served as a negative control and was incubated with a mixture of mAbs 5F8, 5G10, and 6A5.

that post-translational modifications or structural differences in the Lai strain impact the accessibility or stability of these epitopes in native conditions. The ~48 kDa band observed in Fig. 4B likely corresponds to LA0591. Although we cannot fully exclude the possibility of cross-reactivity with other truncated VMPs of similar size, the antibody used was generated against LA0591, and its specificity was supported by control experiments using recombinant protein.

## Anti-VM protein mAbs recognize linear epitopes

Epitope characterization is based on the functional binding of mAbs to antigens or their derivative peptides, which helps identify the position of the epitopes on the antigen (Fig. 4A). Direct ELISA demonstrated that mAbs 5F8 and 5G10 both recognize linear peptides 19 (95–106 aa, NSHGPLQGGGYF) and 20 (99–110 aa, PLQGGGYFFNTA), and they both had overlapping binding affinities. In contrast, peptide 67 (287–298 aa, NRRGSGGYPTSA) showed binding affinity to mAb 6A5 (Fig. 4B). A broader range of peptide reactivity was observed in inbred hamsters vaccinated with full-length LA1402 + ΔLA0591 (Fig. 4C). ΔLA0591 is a mutant of LA0591 in which the active amino acid residues have been altered. The 3D models of LA0591 and LA3490 highlight three conserved immune-reactive regions—Epitope 19 (red), Epitope 20 (blue), and Epitope 67 (green) (Fig. 4D and E). These regions are marked and shown on both front and rotated views of the protein surfaces, helping visualize where these epitopes are located and how accessible they are to the immune system.

## Epitopes are highly conserved in pathogenic *Leptospira*

The epitope peptide sequences 19 (95–106 aa, NSHGPLQGGGYF), 20 (99–110 aa, PLQGGGYFFNTA), and 67 (287–298 aa, NRRGSGGYPTSA), recognized by the 5F8, 5G10, and 6A5 monoclonal antibodies, were aligned with VM paralogs and orthologs encoded by *L. interrogans* serovars Copenhageni and Lai, and *L. borgpetersenii* serovars. All three epitope peptides were highly conserved across the serovars and species (Fig. 5). Epitope peptide 19 showed 100%–5% conservation in Copenhageni proteins LIC12985, LIC12340, LIC12963, LIC10778, and LIC12986, as well as in Lai orthologs LA0591, LA0620, LA3388, and LA0589, whereas its orthologs in *L. borgpetersenii* had less than 75% conservation. Interestingly, epitope peptide 20 was highly conserved (>75%) across all paralogs and orthologs of VM exotoxins within the serovars and species studied. Epitope peptide 67, located at the C-terminal, showed 100%–75% conservation in Copenhageni LIC12985, LIC10695, LIC12340, LIC12844, and LIC10639, as well as in Lai orthologs LA0591, LA3490, LA1400, and LA0589.

However, conservation was below 75% in *L. borgpetersenii* (Fig. 5).

## Monoclonal antibody-based detection of VM proteins in experimentally infected hamsters' serum and urine

Hamsters infected with *L. interrogans* serovar Copenhageni strain Fiocruz L1-130 showed visible illness and gross pathological changes in major organs compared with controls. By day 4 post-infection, infected animals showed clear signs of illness, including jaundice, pulmonary hemorrhage, enlarged and congested liver and spleen, swollen kidneys, and pale abdominal cavities (Fig. 6A). These findings indicate the presence of a systemic infection causing widespread organ involvement.

We retrospectively evaluated the performance of a candidate VM protein-based antigen detection method using these experimentally infected hamster serum and urine samples.

A standard curve was generated using the recombinant LA0591 VM antigen in the capture ELISA to interpolate the VM protein concentration in experimental samples (Fig. 6B and C). VM proteins were detected in both serum and urine from experimentally infected hamsters (Fig. 6D and E). Serum from hamsters infected with different doses of *Leptospira* showed varying VMP levels, ranging from 0.74 ng/mL to 4.4 ng/ml (Fig. 6D).

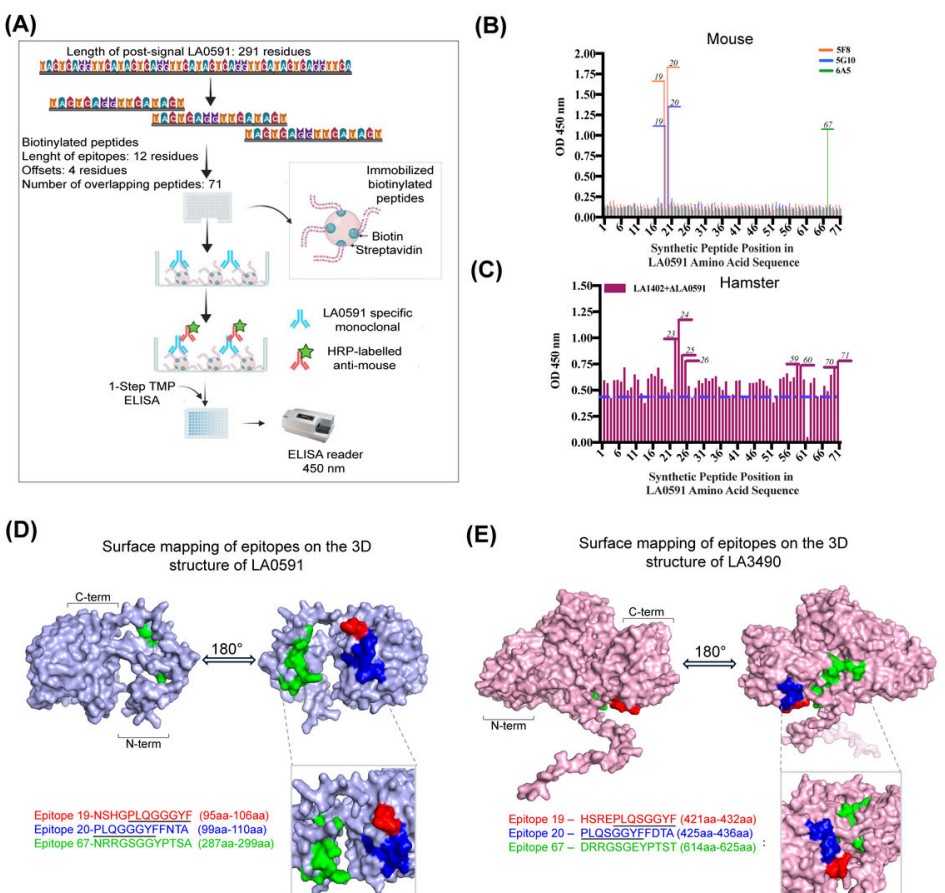

**FIG 4** Epitope mapping of linear B-cell epitopes on the 3D structures of VM proteins (LA0591 and LA3490). This figure illustrates the identification and spatial visualization of linear B-cell epitopes on the leptospiral VM proteins LA0591 and LA3490. (A) A set of 71 biotinylated 12-mer peptides, designed with a 4-residue overlap and spanning the post-signal sequence of the 291-amino-acid LA0591 protein, was synthesized and immobilized on streptavidin-coated 96-well plates for epitope screening. (B) Mapping was performed using monoclonal antibodies (5F8, 5G10, and 6A5) raised in mice, with sera diluted 1:1,000 and detection via anti-mouse IgG-HRP conjugates (5F8 isotype: IgG1; 6A5 isotype: IgG2β) at a 1:10,000 dilution. (C) Additional sera from hamsters vaccinated with LA1402 and ΔLA0591 revealed broader epitope reactivity patterns. The mapped epitopes were further localized on 3D structural models of LA0591 (D) and LA3490 (E), highlighting three conserved immunoreactive regions: Epitope 19 (red), Epitope 20 (blue), and Epitope 67 (green). LA0591 is shown in light blue, and LA3490 in pink, with epitopes displayed on both front and rotated surface views, emphasizing their spatial distribution and surface exposure of immunoreactive epitopes on the modeled structures of VM proteins. Underlined sequences indicate overlapping regions across the identified linear epitopes. The ELISA protocol illustration was created using BioRender.com.

VM proteins were detected and quantified in the urine of hamsters infected with $2 \times 10^8$/mL of *Leptospira*, with concentrations varying from 0.35 ng/mL to 13.4 ng/ml on days 2, 4, and 5 (Fig. 6E).

## DISCUSSION

Here, we demonstrate that *Leptospira*-secreted protein exotoxins, members of the PF07598 gene family encoded virulence-modified proteins (VM proteins), can be detected in the blood and urine of experimentally infected hamsters. To our knowledge, this is the first report of a secreted protein exotoxin being detected in both the blood and urine during systemic *Leptospira* infection, highlighting its potential involvement in disease pathogenesis. Although secreted toxins from other bacterial pathogens, such as anthrax and listeriolysin, have previously been detected in infected animals (32, 33), this represents the first such observation in leptospirosis. This finding expands our

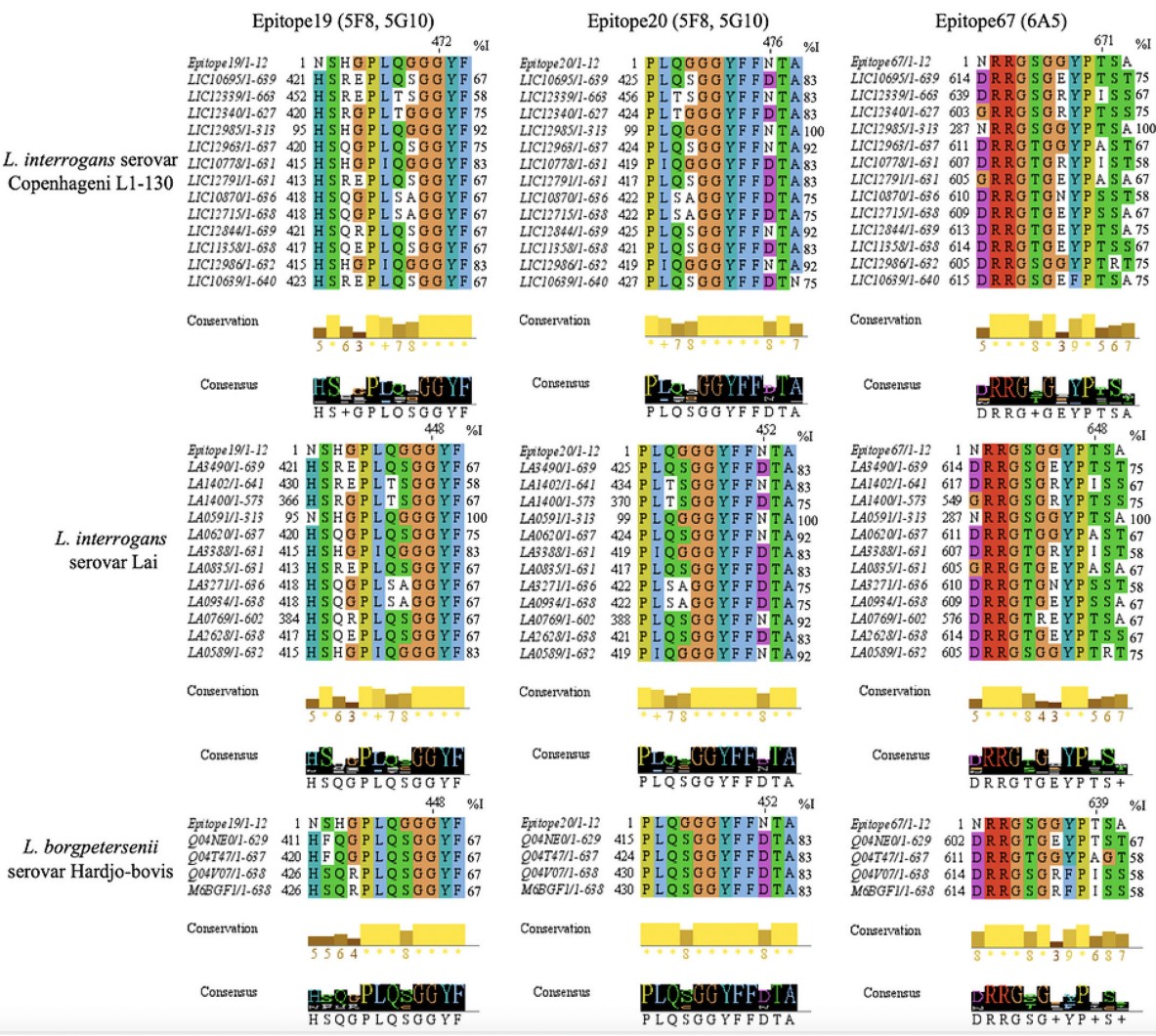

**FIG 5** Epitope-based multiple sequence alignment of the PF07598 gene family encoding VMP exotoxins. Epitope 19 (NSHGPLQGGGYF), Epitope 20 (PLQGGGYFFNTA), and Epitope 67 (NRRGSGGYPTSA) from LA0591 (an ortholog of LIC12985, encoding only the C-terminal and naturally lacking RBL1 and RBL2) were aligned with paralogous VM proteins encoded by *L. interrogans* serovars Lai (*n* = 12), serovar Copenhageni (*n* = 13), and *L. borgpetersenii* serovar Hardjo-bovis encoding Q04NE0, Q04T47, Q04V07, and M6BGE1. These sequences were sequentially aligned using pairwise MUSCLE protein alignment with default settings in Jalview V2.11.4. Percentage identity was determined through pairwise alignment in Jalview V2.11.4. The consensus reflects the abundance of amino acids at each position, whereas conservation is quantified as a numerical index that represents the preservation of physicochemical properties across the alignment.

understanding of leptospiral virulence mechanisms and offers new insight into host–pathogen interactions in this neglected tropical disease.

However, the application of VMP detection as a diagnostic tool remains to be validated and was not addressed in this study. We developed a monoclonal antibody-based immunoassay for circulating VM proteins in blood and urine at 4 days post-infection, a time point that corresponds with the onset of clinical signs in the hamsters experimentally infected with pathogenic *Leptospira*. The 6A5/5F8 monoclonal antibody pair (capturing/detecting) successfully detects and quantifies VM proteins in urine (0.35–13.4 ng/mL) as early as day 2 post-infection with *L. interrogans* Copenhageni strain LI-130, and in serum (0.74–4.4 ng/mL) by day 4, before symptom onset.

Upregulation of VM proteins in human-infecting lethal *Leptospira interrogans* serovar Copenhageni strain L1-130 implies a potential role in severe human leptospirosis and their use as therapeutic targets. Epitope mapping revealed that the linear peptide

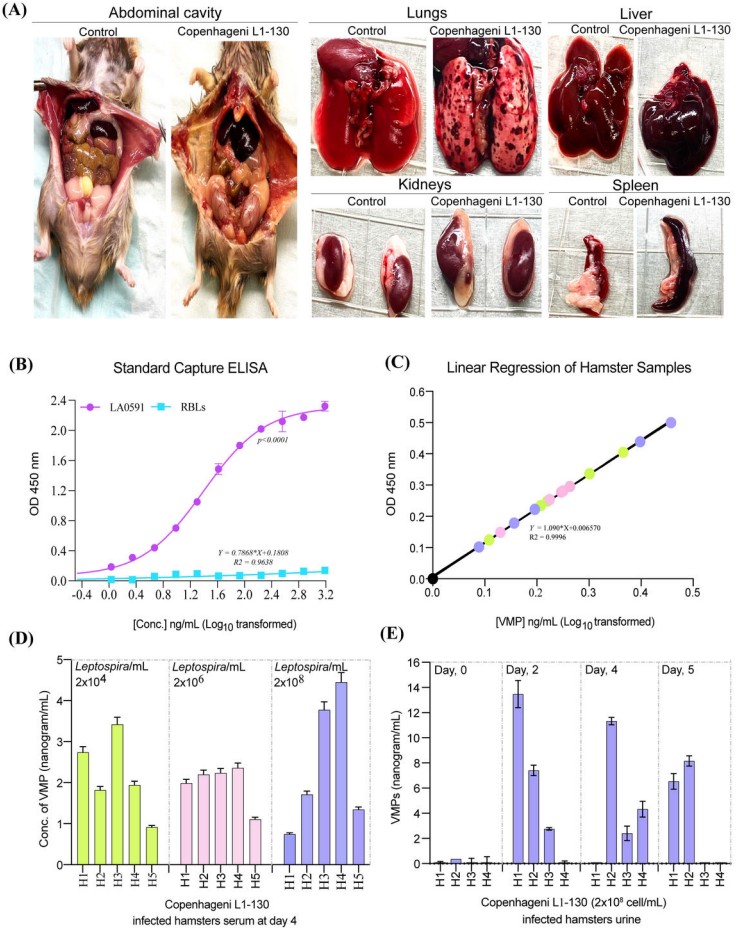

**FIG 6** Capture ELISA-based detection of VM antigen and associated pathology in *L. interrogans* serovar Copenhageni strain Fiocruz-L1-130-infected hamsters. (A) Gross pathology of major organs such as the abdominal cavity, lungs, liver, kidneys, and spleen in control and *L. interrogans* serovar Copenhageni serovar Fiocruz strain L1-130-infected ($2 \times 10^8$ leptospires/mL, day 4) hamster. Infected animals show marked pathological changes, including jaundice and hemorrhagic lungs, congested and enlarged liver and spleen, swollen kidneys, and pale abdominal cavities, indicating systemic infection and organ damage. (B) A standard ELISA was performed using two antibody combinations (capturing 6A5: 1:1,000 and biotinylated detecting 5F8: 1:1,000) to detect LA0591, with RBLs as the negative control. (C) Linear regression was generated using the ELISA data from (B) with recombinant LA0591. The resulting equation is $Y = 1.090*X + 0.006570$ with R2 = 0.9996. (D) Hamsters infected with varying doses ($2 \times 10^4$, $2 \times 10^6$, or $2 \times 10^8$) of *L. interrogans* serovar Copenhageni strain Fiocruz-L1-130, P1 culture had blood collected on day 4. Serum (1:50 dilution) was tested for VM antigen using the optimized capture ELISA. (E) Hamster urine samples from those infected with $2 \times 10^8$ leptospires/mL of serovar Copenhageni L1-130 were collected on days 0, 2, 4, and 5 and analyzed for VM antigen excretion. A standard ELISA curve was used to interpolate VM antigen concentrations in hamster samples. The same analysis was performed on samples from non-infected hamsters as negative controls. The values obtained from these controls were subtracted and accounted for in the representative figure to ensure accurate interpretation. The results represent the mean ± SD for triplicate experiments.

epitopes are highly conserved across pathogenic *Leptospira* species and serovars, underlining their potential for vaccine development and therapeutic interventions. These results support the rapid development of a lateral flow assay, which could significantly improve diagnostic capabilities, particularly in resource-limited settings.

Previous meta-analyses have provided quantitative estimates of *Leptospira* spp. loads in various hosts, including rats ($5.7 \times 10^6$ /mL urine), mice ($3.1 \times 10^3$ /mL), cattle (3.7

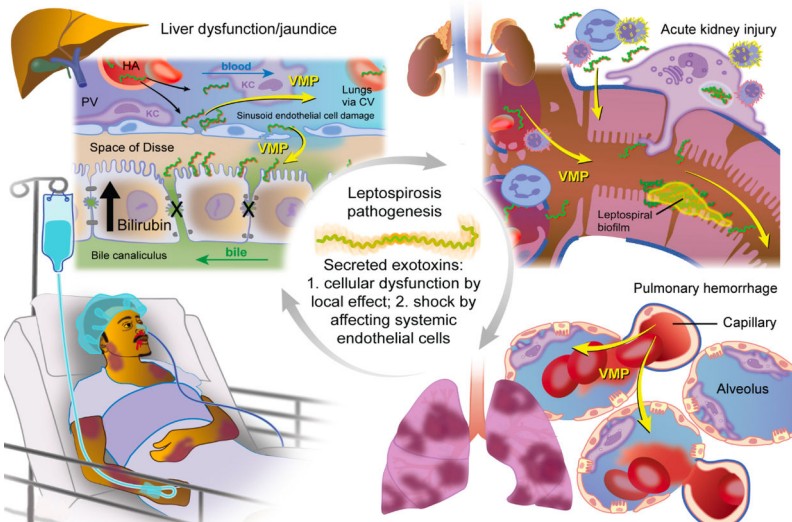

**FIG 7** Hypothetical schematic representation of VM proteins in mediated leptospirosis pathogenesis. The underlying hypothesis is that systemically circulating leptospiral VM proteins mediate/contribute to multiple organ dysfunction, particularly liver (jaundice), kidney (acute kidney injury), and lung (hemorrhage). Liver dysfunction, where they damage sinusoidal endothelial cells, disrupting bile flow and causing bilirubin buildup—a key factor behind jaundice. In the kidneys, *Leptospira* organisms colonize the proximal renal tubules, form biofilms, and release VMPs that compromise endothelial integrity, ultimately contributing to acute kidney injury. Within the lungs, infection leads to capillary damage driven by VMP exotoxins, resulting in blood leakage into the alveolar spaces and subsequent pulmonary hemorrhage. Clinically, patients with leptospirosis often present with a combination of jaundice, renal impairment, and pulmonary complications. These manifestations are associated with systemic infection and may involve either primary or secondary action of VMP exotoxins, although their direct role in mediating tissue damage remains to be fully elucidated.

$\times 10^4$ /mL), and humans ($7.9 \times 10^2$ /mL) (35–37). However, differences in sample size, detection methods, and *Leptospira* species and serovars limit cross-species comparisons. The presence of *Leptospira* in the kidney often reflects levels in urine. Supporting this observation, Costa et al. observed a strong correlation between kidney and urinary leptospiral loads in brown rats using qPCR. In humans, reported urinary leptospiral loads vary widely, from ~$7.9 \times 10^2$ to $5.7 \times 10^6$ /mL, due to factors such as infection stage, host response, and detection methodology (35–37).

Pathogenic *Leptospira* is present in the bloodstream (leptospiremia) up to 14 days of fever prior to antibody formation and clearance from the blood (8–10, 13). The bacteria subsequently colonize the renal tubules and are excreted in urine, with urinary shedding of leptospires (>$10^6$/mL) and leptospiral antigens/secretory protein or toxins serving as key diagnostic markers (38, 39). Leptospiral antigens have also been detected in tissue sections from infected animals and humans, implicating their role in pathogenesis (40–43). These antigens are valuable for assessing renal injury across species, including domestic dogs, cats, livestock, and humans (31, 37, 44, 45). The detection of leptospiral antigens/toxins offers a reliable and early diagnostic approach for bacterial infections, presenting advantages over serology, culture, and PCR in terms of accuracy, efficiency, and cost-effectiveness. This method is particularly appealing due to its high specificity, strong correlation with disease pathogenesis, and independence from host immune responses. As such, it serves as a valuable tool for accurately assessing disease burden and gaining deeper insights into disease mechanisms. In our ongoing efforts, we are broadening the specificity evaluation of the assay by testing the selected mAbs against antigens from other relevant pathogens, as well as samples from diverse geographical settings. This will allow us to assess potential cross-reactivity and further establish the diagnostic specificity of the capture ELISA.

Pathogenic bacteria have evolved various virulence factors and toxins that facilitate host-pathogen interactions and contribute to tissue damage. Although the clinical and pathological aspects of leptospirosis are well-characterized, the role of leptospiral toxins in disease pathogenesis remains limited until the identification of the PF07598 gene family encoding VM protein exotoxins in pathogenic *Leptospira*. The absence of this gene family in non-pathogenic strains underscores its significance as a virulence factor (11, 19).

VM protein levels are likely to correlate more closely with the actual *Leptospira* burden in blood or urine than with the initial inoculum dose. This hypothesis is supported by the observed variability in VM levels among hamsters within the same infection group. Although the present study primarily aimed to establish the presence of VM proteins using a capture ELISA, future work will focus on quantifying *Leptospira* load in parallel with VM protein detection. This approach will enable a more accurate assessment of the assay's sensitivity, particularly in low-dose infections, and provide a clearer understanding of the relationship between toxin production and bacterial burden during infection.

Within spirochetes, the PF07598 gene family encoding VM proteins is unique to *Leptospira*, based on a comprehensive GenBank analysis. Interestingly, homologs of this gene family are present in unrelated α-proteobacteria, including *Bartonella bacilliformis*, *Bartonella ancashi*, and *B. australis*, each with multiple paralogs. Additionally, single gene copies are found in a few ε-proteobacteria species such as *Helicobacter hepaticus*, *H. mustelae*, and *H. cetorum* (46). Although *B. bacilliformis* and *B. ancashi* infect humans, they do not overlap clinically or epidemiologically with *Leptospira*. Cross-reactivity with these proteobacteria presents a potential avenue for future investigation.

Our discovery of leptospiral VMPs has led to the hypothesis that these proteins function as secreted exotoxins contributing to severe and fatal leptospirosis.

At the cellular level, VMPs may directly or indirectly target critical organs such as the lungs, liver, and kidneys. In the lungs, VMPs could disrupt tight junction proteins in endothelial cells, potentially leading to pulmonary hemorrhage. In the kidneys, leptospires may colonize and form biofilms, with secretion of VMPs exotoxins enabling proximal tubule colonization and potentially contributing to acute kidney injury. Similarly, in the liver, VMP-mediated effects may contribute to hepatic dysfunction (Fig. 7). Future work will focus on the identification of specific target cells and receptor-ligand interactions involved in VM protein binding, which is fundamental to understanding the mechanistic basis of the clinical pathogenesis of leptospirosis.

Whole genome sequencing has identified virulence factors such as sphingomyelinases, collagenase, pore-forming toxins, hemolysins, ricin B-like cytotoxins (VM protein), lipopolysaccharides, and others, with a potential role in leptospirosis pathogenesis and diagnosis (47–50). Although ELISA-based diagnostics offer advantages over the MAT, the sensitivity and specificity of whole bacterial lysate remain limited in the early phase due to the delayed immune response against the O-antigen polysaccharide (16, 51–53). Leveraging the secreted or shed antigens in urine or blood for early detection and treatment of leptospirosis can facilitate rapid recovery and prevent progression to severe leptospirosis.

Bacterial toxins, such as those from *B. anthracis* (Edema toxin, ET), *B. pertussis* (Pertussis toxin, PT), and *C. botulinum* (Botulinum neurotoxin, BoNT), are routinely diagnosed (54). Over the past decade, various methods, including HPLC, HPLC-MS, MALDI-TOF, ELISA, PCR, and fluorescence assays, have been developed for sensitive and selective toxin detection (55, 56). These methods are valuable for identifying unknown toxins. Recent advancements in antibody engineering and nanomaterials (eg. AuNPs, MNPs, QDs, and MOFs) have further improved detection by enhancing separation, recognition, and signal amplification, offering promising solutions for better analytical performance. Antibodies are widely used for the rapid diagnosis and therapy of human diseases due to their high affinity and specificity for target molecules (55, 57, 58). Antibody-mediated epitope mapping facilitates the development of multi-epitope vaccines. Epitopes, short amino acid sequences, trigger a more potent immune response than the full protein (59). Epitope-based vaccines offer several advantages, such as

faster design, cost-effective formulations, and improved immunogenicity with fewer side effects, as supported by *in vitro* and *in vivo* studies (60).

The current study has the potential to both advance antigen-based early diagnosis of leptospirosis and develop a multi-epitope-based vaccine. A limitation of this study is that although the mAbs effectively detect LA0591 in ELISA, their detection of full-length recombinant VM proteins is significantly weaker, which may hinder the detection of full-length VM proteins in experimental samples. A limitation of this study is the absence of basic clinical tests, such as the complete blood count and blood chemistries, such as the leukocyte and platelet counts, and transaminases, bilirubin, blood urea nitrogen, creatinine, and bilirubin, which are important indicators of leptospirosis severity and organ dysfunction.

We do not have information on VM protein kinetics in blood or urine at earlier time points after the challenge infection. It is possible that VM protein levels may remain below the detection threshold of our capture ELISA, and theoretically, the presence of VM protein levels in urine following high-dose infection may result from spillover due to bacteremia or bacterial lysis rather than active secretion by *Leptospira* in the renal environment. Our ongoing studies are tracking VM protein kinetics in blood and urine throughout infection, particularly during the later stages. These investigations will include analysis of kidney tissue to assess local VM expression and its association with tissue pathology and abnormalities of urine and blood kidney function tests. Such studies will help distinguish between passive leakage and active secretion of VM proteins, ultimately refining their potential as diagnostic markers for both acute and chronic leptospirosis. We plan to test and validate the diagnostic utility of VM protein antigen detection in blood and urine in diverse settings of human leptospirosis.

## ACKNOWLEDGMENTS

This work was supported by the United States Public Health Service through the National Institutes of Health, NIAID grants R01AI108276, U19AI115658, 1R41AI174377, and 1R41AI181135; and by the America's Foundation.

Conceptualization: R.C. and J.M.V., Data curation: R.C., A.J., and J.T., formal analysis: R.C., A.J., and J.T., Funding acquisition: J.M.V., investigation: R.C., A.J., J.T., and J.M.V., methodology: R.C., A.S., J.T., S.D., and J.M.V., resources: J.M.V., Supervision: R.C. and J.M.V., Visualization: R.C., J.T., S.D., and J.M.V., Writing - original draft: R.C., and Writing - review and editing: R.C., J.T., S.D., and J.M.V.

## AUTHOR AFFILIATIONS

[1]Section of Infectious Diseases, Department of Internal Medicine, Yale University School of Medicine, New Haven, Connecticut, USA
[2]Luna Bioscience Inc., New Haven, Connecticut, USA

## AUTHOR ORCIDs

Joseph M. Vinetz  http://orcid.org/0000-0001-8344-2004

## FUNDING

| Funder | Grant(s) | Author(s) |
| --- | --- | --- |
| National Institute of Allergy and Infectious Diseases | R01AI108276, U19AI115658, 1R41AI174377, 1R41AI181135 | Joseph M. Vinetz |
| America's Foundation | 001 | Joseph M. Vinetz |

## AUTHOR CONTRIBUTIONS

Reetika Chaurasia, Conceptualization, Data curation, Formal analysis, Investigation, Methodology, Writing – original draft, Visualization | Andrea Jacobs, Data curation, Investigation, Methodology | Jie Tang, Data curation, Investigation, Methodology | Songyu Dong, Investigation, Methodology | Joseph M. Vinetz, Conceptualization, Data curation, Formal analysis, Funding acquisition, Investigation, Methodology, Project administration, Validation, Writing – original draft, Writing – review and editing

## ADDITIONAL FILES

The following material is available online.

### Supplemental Material

**Fig. S1 (Spectrum00018-25-s0001.tiff).** Purity and immunoreactivity of rVM proteins.
**Fig. S2 (Spectrum00018-25-s0002.tif).** Clinical progression in hamsters following *L. interrogans* serovar Copenhageni strain Fiocruz L1-130 infection at the time points.
**Supplemental material (Spectrum00018-25-s0003.docx).** Supplemental figure captions.

### Open Peer Review

**PEER REVIEW HISTORY (review-history.pdf).** An accounting of the reviewer comments and feedback.

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
