## [Reviewer comments · Microbiology Spectrum]

Microbiology Spectrum

Development of Leptospiral Virulence-Modifying Protein Detection Assay: Implications for Pathogenesis and Diagnostic Test Development

Reetika Chaurasia, Andrea Jacobs, Jie Tang, Songyu Dong, and Joseph Vinetz

Corresponding Author(s): Joseph Vinetz, Yale School of Medicine

Review Timeline:

Submission Date:	January 3, 2025
Editorial Decision:	April 18, 2025
Revision Received:	May 22, 2025
Editorial Decision:	July 2, 2025
Revision Received:	August 5, 2025
Accepted:	August 19, 2025

Editor: Denis Sereno

Reviewer(s): Disclosure of reviewer identity is with reference to reviewer comments included in decision letter(s). The following individuals involved in review of your submission have agreed to reveal their identity: Amaro Nunes Duarte-Neto (Reviewer #4)

Transaction Report:

DOI: <https://doi.org/10.1128/spectrum.00018-25>

Re: Spectrum00018-25 (Development of Leptospiral Virulence-Modifying Protein Detection Assay: Implications for Pathogenesis and Diagnostic Test Development)

Dear Prof. Joseph M. Vinetz:

Thank you for the privilege of reviewing your work. Below you will find my comments, instructions from the Spectrum editorial office, and the reviewer comments.

Revision Guidelines

Sincerely,
Denis Sereno
Editor
Microbiology Spectrum

Reviewer #2 (Comments for the Author):

The study developed VM protein antibodies, and compared interactions of different antibody isotypes with VM exotoxin antigens. The study was continuous work from previous publications.

1. Descriptions in this manuscript and its references showed that VM exotoxin associates with Leptospiral virulence, but the

reviewer was not convinced of the causality according to Koch's Postulates. The reviewer did not identify the information that VM exotoxin-knockout *Leptospira* lost or had attenuated virulence, nor VM exotoxin-overexpressing *Leptospira* exhibited increased virulence. The virulence mechanism of VM exotoxin was not experimentally proved.

2. Do other *Sprichaeetes* species harbor VM toxin homo or paralogs? If so, is there cross-reactivity of mAbs developed in this study against other *Sprichaeetes*?
3. Please provide evidence that tests in the manuscript actually studied extracellularly-secreted VM exotoxins, and whether VM exotoxin does or does not exist intracellularly. Would dead or lysed bacterial cells interfere with the tests? Do the antibodies interact with cell pellets? In hamster assays (Figure 6), were extracellular, intracellular and from lysed/dead cells VM exotoxins differentiated?
4. Not quite understand the selection process of these four antibody isotypes. For instance, four antibody isotypes were determined to have high affinity against the antigen, but only three combinations were selected in Capture-ELISA. Second, why 5E10 was not evaluated? Especially 6A5/5E10 pairing showed quite high affinity in Table 2. Please rephrase the selection process and if needed, reorganize figures and result sections.
5. In Figure 6, What is the meaning of "H1~H5", total 5 hamsters? On Day 4 with 2×10^8 cells/mL infection dose, VM toxin levels in serum and urine were not proportional for H2, H3, and H4. What is the mechanism behind?
6. SDS-PAGE results in Figure 3 were in poor quality, and even bands in markers were not straight or clear. Publication needs better quality.
7. Please review the manuscript text, correct typos and clarify/improve phrasing. For instance, in Table 1, correct terms in right upper and lower case forms. Secondly, what is the unit in Table 2? The high, the better? Suggest to add necessary description in Methods or table caption.

Reviewer #3 (Comments for the Author):

Comments on the manuscript titled "Development of Leptospiral Virulence-Modifying Protein Detection Assay: Implications for Pathogenesis and Diagnostic Test Development"

In this study, the authors developed a monoclonal antibody (mAb)-based capture immunoassay to detect VM exotoxins in experimental hamster models, thereby validating the hypothesis that VM proteins function as secretory exotoxins. Monoclonal antibodies were generated against a natural variant LA0591, a VM protein containing a conserved C-terminal DNase toxin domain. Using mAbs 6A5 and 5F8, a capture ELISA successfully detected circulating VM exotoxins in serum and urine from infected hamsters, confirming the secretion of these proteins during infection.

The comments on the role of VM toxins in diagnosis and pathogenesis are discussed separately as follows:

I. Diagnosis

1. Line 321-323: "In the indirect ELISA, two-fold serial dilutions of 5F8, 6A5, and 5G10 mAbs were tested against a fixed amount of antigen, resulting in detection limits of 1.98 ng/mL for 5F8, 8.56 ng/mL for 6A5, and 0.33 μ g/mL for 5G10 (Fig. 2A)." Since a fixed amount of antigen was used, it is unclear how this experiment was designed to determine and calculate the LoD. Please clarify the methodology used for LoD estimation.
2. Line 326-329: "The two selected pairs, 6A5/5F8 and 5F8/6A5, were further evaluated using two-fold serial dilutions of LA0591 (starting at 1,000 ng/mL), with detection limits of 1.84 ng/mL for 6A5/5F8 and 1.31 ng/mL for 5F8/6A5 (Fig. 2C,D)." Based on the results shown in Fig. 2C and 2D, the LoD of the 6A5/5F8 pair would be expected to be better (i.e., lower) than that of 5F8/6A5. Please explain why the reported LoD of 6A5/5F8 was higher than that of 5F8/6A5.
3. The results of epitope mapping in Figure 5 should be visualized on the 3D structure shown in Figure 1. Highlighting the epitopes recognized by each mAb will provide clearer insight and help correlate with the epitope binning results presented in Table 2.
4. In Figure 3A, the expected molecular weights of paralogous VTMs should be mentioned to aid in interpreting the results. Including the SDS-PAGE images corresponding to the Western blots (in the Supplement) would help confirm that the absence of bands was not due to undetectable or low protein levels. Furthermore, it is unclear why 6A5 recognizes a single band of LA0591, while other mAbs detect multiple bands, despite using the same recombinant protein. Comparing the epitope sequences recognized by each mAb to those in other paralogous VTMs, as done in Figure 5, would help clarify differences in specificity and cross-reactivity—particularly why mAb 6A5 exhibited broader reactivity.
5. According to Figure 5, the conserved epitopes recognized by mAbs 5F8/5G10 and 6A5 should allow detection of VTM in the Lai strain, especially under induced conditions (Figure 3B). Please clarify why little or no reactivity was observed. Additionally, since the mAbs detected targets under denaturing conditions in SDS-PAGE/Western blot, they are likely to recognize linear epitopes.
6. Although the mAbs demonstrated diagnostic potential in hamsters, the limitations of the study should be addressed. While the capture ELISA detected VM antigens in blood and urine samples from hamsters, it remains unclear whether the assay is sufficiently sensitive for use with human clinical samples. Can it achieve the necessary LoD to detect VM proteins in clinical

settings? What is the expected concentration of VM antigens in human samples, and how does this correlate with leptospiral burden? These points should be discussed with reference to published data on leptospiral loads in human blood and urine. Furthermore, the specificity of the assay has not been assessed against antigens from other pathogens. Potential cross-reactivity should be evaluated to confirm the diagnostic specificity of the capture ELISA using the selected mAbs.

II. Pathogenesis

To support the conclusion that "The presence of leptospiral VM exotoxins in the serum and urine of infected hamsters supports the hypothesis that circulating VM exotoxins mediate disease pathogenesis," it is necessary to correlate VM toxin levels with disease severity in infected hamsters or humans.

1. In this study, capture ELISA was used to detect and quantify VM exotoxins in the serum and urine of hamsters (Fig. 6C, D) infected with different inoculum doses of *Leptospira*. However, VM protein levels should more accurately correlate with the actual number of *Leptospira* present in the blood or urine rather than with the inoculum dose alone, as toxin levels varied among hamsters within the same group. Furthermore, since a direct correlation between *Leptospira* load and VM protein levels was not assessed, it remains unclear whether the assay is sufficiently sensitive to detect VM exotoxins at lower inoculum doses.
2. Were VM exotoxin levels significantly different among hamsters infected with varying inoculum doses? In Figure 6C, serum toxin levels do not appear to be clearly dose-dependent. The study would be strengthened by correlating inoculum dose and VM levels with severity outcomes, such as survival and organ pathology, to demonstrate that higher VM levels are associated with more severe disease.
3. Please explain why VM toxin levels were measured only on Days 0-5 in the high-dose group (2×10^8 cells/mL), as shown in Figure 6D. During the early stages of infection, *Leptospira* are expected to be more abundant in the blood than in the urine. The authors did not provide data on survival or disease severity in the infected hamsters. In lower-dose infections, hamsters may survive longer and progress into the leptospiruric phase, during which urinary shedding of VM proteins could be evaluated. At early time points, VM toxin levels may be too low for detection by capture ELISA, and their presence in urine following high-dose infection may reflect spillover from bacteremia or bacterial lysis rather than active secretion of VM toxins by *Leptospira* in the kidneys. Renal tissues should be analyzed for VM protein and correlated with histopathological findings.
4. In light of the above points, the detection of VM toxins in the blood and urine of infected hamsters-based on capture ELISA- and the absence of detection in the lung may be overstated in the summary presented in Figure 7. Please revise the interpretation accordingly.

Reviewer #4 (Comments for the Author):

The study by Chaurasia et al. adds a new and important mechanism to the pathogenesis of leptospirosis by providing the first evidence of secreted leptospiral exotoxins in the bloodstream of infected animals, and establishes a basis for the development of novel diagnostic and therapeutic approaches for the disease. The authors concluded that further work is needed to investigate the tissue-specific expression of VM proteins.

Although the authors have not demonstrated the presence of MV proteins in the tissues, I would like to suggest that they describe the clinical condition of the hamsters throughout the course of the disease and at the time of euthanasia in the results, and if they could include graphs showing the kinetics of MV proteins in the blood and urine throughout the experiment, associated with the clinical condition of the animals during the experiment. I suggest that the authors describe the macroscopic appearance of the lungs, liver, kidneys and spleen at the time of euthanasia, with the possibility of providing a figure showing the appearance of the organs when the thoracic and abdominal cavities are opened. Is there any data on leukocyte count, urea, creatinine or bilirubin during the experiments.

The study developed VM protein antibodies, and compared interactions of different antibody isotypes with VM exotoxin antigens. The study was continuous work from previous publications.

1. Descriptions in this manuscript and its references showed that VM exotoxin associates with Leptospiral virulence, but the reviewer was not convinced of the causality according to Koch's Postulates. The reviewer did not identify the information that VM exotoxin-knockout *Leptospira* lost or had attenuated virulence, nor VM exotoxin-overexpressing *Leptospira* exhibited increased virulence. The virulence mechanism of VM exotoxin was not experimentally proved.
2. Do other *Sprichaeetes* species harbor VM toxin homo or paralogs? If so, is there cross-reactivity of mAbs developed in this study against other *Sprichaeetes*?
3. Please provide evidence that tests in the manuscript actually studied extracellularly-secreted VM exotoxins, and whether VM exotoxin does or does not exist intracellularly. Would dead or lysed bacterial cells interfere with the tests? Do the antibodies interact with cell pellets? In hamster assays (Figure 6), were extracellular, intracellular and from lysed/dead cells VM exotoxins differentiated?
4. Not quite understand the selection process of these four antibody isotypes. For instance, four antibody isotypes were determined to have high affinity against the antigen, but only three combinations were selected in Capture-ELISA. Second, why 5E10 was not evaluated? Especially 6A5/5E10 pairing showed quite high affinity in

Table 2. Please rephrase the selection process and if needed, reorganize figures and result sections.

5. In Figure 6, What is the meaning of "H1~H5", total 5 hamsters? On Day 4 with 2×10^8 cells/mL infection dose, VM toxin levels in serum and urine were not proportional for H2, H3, and H4. What is the mechanism behind?
6. SDS-PAGE results in Figure 3 were in poor quality, and even bands in markers were not straight or clear. Publication needs better quality.
7. Please review the manuscript text, correct typos and clarify/improve phrasing. For instance, in Table 1, correct terms in right upper and lower case forms. Secondly, what is the unit in Table 2? The high, the better? Suggest to add necessary description in Methods or table caption.

Comments on the manuscript titled “Development of Leptospiral Virulence-Modifying Protein Detection Assay: Implications for Pathogenesis and Diagnostic Test Development”

In this study, the authors developed a monoclonal antibody (mAb)-based capture immunoassay to detect VM exotoxins in experimental hamster models, thereby validating the hypothesis that VM proteins function as secretory exotoxins. Monoclonal antibodies were generated against a natural variant LA0591, a VM protein containing a conserved C-terminal DNase toxin domain. Using mAbs 6A5 and 5F8, a capture ELISA successfully detected circulating VM exotoxins in serum and urine from infected hamsters, confirming the secretion of these proteins during infection.

The comments on the role of VM toxins in diagnosis and pathogenesis are discussed separately as follows:

I. Diagnosis

1. Line 321–323: “In the indirect ELISA, two-fold serial dilutions of 5F8, 6A5, and 5G10 mAbs were tested against a fixed amount of antigen, resulting in detection limits of 1.98 ng/mL for 5F8, 8.56 ng/mL for 6A5, and 0.33 µg/mL for 5G10 (Fig. 2A).”
Since a fixed amount of antigen was used, it is unclear how this experiment was designed to determine and calculate the LoD. Please clarify the methodology used for LoD estimation.
2. Line 326–329: “The two selected pairs, 6A5/5F8 and 5F8/6A5, were further evaluated using two-fold serial dilutions of LA0591 (starting at 1,000 ng/mL), with detection limits of 1.84 ng/mL for 6A5/5F8 and 1.31 ng/mL for 5F8/6A5 (Fig. 2C,D).”
Based on the results shown in Fig. 2C and 2D, the LoD of the 6A5/5F8 pair would be expected to be better (i.e., lower) than that of 5F8/6A5. Please explain why the reported LoD of 6A5/5F8 was higher than that of 5F8/6A5.
3. The results of epitope mapping in Figure 5 should be visualized on the 3D structure shown in Figure 1. Highlighting the epitopes recognized by each mAb will provide clearer insight and help correlate with the epitope binning results presented in Table 2.
4. In Figure 3A, the expected molecular weights of paralogous VTMs should be mentioned to aid in interpreting the results. Including the SDS-PAGE images corresponding to the Western blots (in the Supplement) would help confirm that the absence of bands was not due to undetectable or low protein levels. Furthermore, it is unclear why 6A5 recognizes a single band of LA0591, while other mAbs detect multiple bands, despite using the same recombinant protein. Comparing the epitope sequences recognized by each mAb to those in other paralogous VTMs, as done in Figure 5, would help clarify differences in specificity and cross-reactivity—particularly why mAb 6A5 exhibited broader reactivity.
5. According to Figure 5, the conserved epitopes recognized by mAbs 5F8/5G10 and 6A5 should allow detection of VTM in the Lai strain, especially under induced conditions (Figure 3B). Please clarify why little or no reactivity was observed. Additionally, since the

mAbs detected targets under denaturing conditions in SDS-PAGE/Western blot, they are likely to recognize linear epitopes.

6. Although the mAbs demonstrated diagnostic potential in hamsters, the limitations of the study should be addressed. While the capture ELISA detected VM antigens in blood and urine samples from hamsters, it remains unclear whether the assay is sufficiently sensitive for use with human clinical samples. Can it achieve the necessary LoD to detect VM proteins in clinical settings? What is the expected concentration of VM antigens in human samples, and how does this correlate with leptospiral burden? These points should be discussed with reference to published data on leptospiral loads in human blood and urine. Furthermore, the specificity of the assay has not been assessed against antigens from other pathogens. Potential cross-reactivity should be evaluated to confirm the diagnostic specificity of the capture ELISA using the selected mAbs.

II. Pathogenesis

To support the conclusion that “The presence of leptospiral VM exotoxins in the serum and urine of infected hamsters supports the hypothesis that circulating VM exotoxins mediate disease pathogenesis,” it is necessary to correlate VM toxin levels with disease severity in infected hamsters or humans.

1. In this study, capture ELISA was used to detect and quantify VM exotoxins in the serum and urine of hamsters (Fig. 6C, D) infected with different inoculum doses of *Leptospira*. However, VM protein levels should more accurately correlate with the actual number of *Leptospira* present in the blood or urine rather than with the inoculum dose alone, as toxin levels varied among hamsters within the same group. Furthermore, since a direct correlation between *Leptospira* load and VM protein levels was not assessed, it remains unclear whether the assay is sufficiently sensitive to detect VM exotoxins at lower inoculum doses.
2. Were VM exotoxin levels significantly different among hamsters infected with varying inoculum doses? In Figure 6C, serum toxin levels do not appear to be clearly dose-dependent. The study would be strengthened by correlating inoculum dose and VM levels with severity outcomes, such as survival and organ pathology, to demonstrate that higher VM levels are associated with more severe disease.
3. Please explain why VM toxin levels were measured only on Days 0–5 in the high-dose group (2×10^8 cells/mL), as shown in Figure 6D. During the early stages of infection, *Leptospira* are expected to be more abundant in the blood than in the urine. The authors did not provide data on survival or disease severity in the infected hamsters. In lower-dose infections, hamsters may survive longer and progress into the leptospiruric phase, during which urinary shedding of VM proteins could be evaluated. At early time points, VM toxin levels may be too low for detection by capture ELISA, and their presence in urine following high-dose infection may reflect spillover from bacteremia or bacterial lysis rather than active secretion of VM toxins by *Leptospira* in the kidneys.

Renal tissues should be analyzed for VM protein and correlated with histopathological findings.

4. In light of the above points, the detection of VM toxins in the blood and urine of infected hamsters—based on capture ELISA—and the absence of detection in the lung may be overstated in the summary presented in Figure 7. Please revise the interpretation accordingly.

May 2025

To the Editor:

Thank you for your decision and the reviews on our manuscript. Just to emphasize, this manuscript reports a very significant achievement both in the leptospirosis field (identifying and measuring a pathogenic toxin in the blood of infected animals) and in bacterial pathogenesis (such detection with direct potential relevance to diagnostics and vaccine development) has not been previously reported. We accept our role in not making sufficiently clear to the Editor and reviewers the work's enormous—really unprecedented—impact on the leptospirosis field our work will have. The reviewers' comments do not reflect our proposed high impact of the central finding: that this is the first time in the leptospirosis field—indeed in any field of bacterial infections—in which a secreted, pathogenetic protein exotoxin can be detected in the blood of an infected animal. This finding has important implications in multiple areas: 1) understanding the molecular and cellular pathogenesis of leptospirosis; 2) being the first to detect a bacterial-secreted protein exotoxin in the blood of an infected animal; and 3) providing the basis for novel diagnostics and vaccine development. All told, we suggest that our work will make important contributions to a neglected field in which each discovery reflects a long-fought battle to obtain data.

As follows, we respond, point-by-point, to the reviewer/editor comments, and all changes have been marked using track changes in the main manuscript. A clean version of the revised manuscript is also attached.

Some of the experiments we intended to pursue remain unfinished, as the first author and other contributing members have transitioned out of the lab. However, these studies are planned to continue in future work.

Reviewer #2 (Comments for the Author):

The study developed VM protein antibodies and compared interactions of different antibody isotypes with VM exotoxin antigens. The study was continuous work from previous publications.

1. Descriptions in this manuscript and its references showed that VM exotoxin associates with Leptospiral virulence, but the reviewer was not convinced of the causality according to Koch's Postulates. The reviewer did not identify the information that VM exotoxin-knockout *Leptospira* lost or had attenuated virulence, nor VM exotoxin-overexpressing *Leptospira* exhibited increased virulence. The virulence mechanism of VM exotoxin was not experimentally proved.

- Thank you for the valuable suggestion. We previously experimentally demonstrated that VMPs are exoproteins, secreted extracellularly, possessing a signal peptide and detectable in the supernatant (cell-free medium) of pathogenic *Leptospira* (1). We published in this paper that VMPs exert cytotoxic activity *in vitro* on HeLa cells. We have also shown cytotoxicity to primary pulmonary endothelial cells *in vitro*, work that continues and for which NIH grant applications have been submitted. The current study provides data that support the hypothesis that VMPs are also secreted into the bloodstream of *Leptospira*-infected hamsters. Our developed capture ELISA assay detects VMPs at picogram per milliliter concentrations. Our new capture ELISA is further promising for detecting VMPs in the blood of patient samples (ongoing study). The present work is essential to publish because our data support the conclusions that we draw, which is the most essential characteristic of papers published in *Microbiology Spectrum*, as stated in this ASM journal's scope, "Rather than making subjective evaluations of potential impact, *Microbiology Spectrum* publishes research studies that are of high technical quality and are useful to the community."
- We showed that the treatment of HeLa cells with rLA3490 (one of the VMPs, highly upregulated in hamster blood at day 4, (2), led to cytoskeleton disassembly, caspase-3 activation, and nuclear fragmentation, and was rapidly cytolethal. The N-terminal are *bona fide* ricin B chain-like lectin domain and C-terminal had DNase activity on mammalian and bacterial plasmid DNA. The combination of cell surface binding, internalization, nuclear

translocation, and DNase functions indicates that LA3490 and other VM proteins evolved as novel forms of the bacterial AB domain-containing toxin paradigm (1)

➤ (Line 81 - 94)

“Mathieu Picardeau and his group recently used CRISPR-dcas9 knockdown of the LIMLP11655 (VMP) in *L. interrogans* serovar Manilae supports the hypothesis (and our previous published work) that VMPs are a key virulence factor in the cellular pathogenesis of leptospirosis (3), which underscores the potential critical role of the PF07598 gene family in leptospiral pathogenesis (2, 4-6). The same group recently used dual RNA-Seq to study how *L. interrogans* affects host and pathogen gene expression during infection. They found that only two of 12 *L. interrogans* serovar Manilae PF07598-encoded VM proteins were expressed at the transcriptional level (*L. interrogans* serovar Manilae nomenclature, *LIMLP_11655* and *LIMLP_11660*; orthologs in *L. interrogans* serovar Lai, *LA1400* and *LA1402*, and serovar Copenhageni *LIC12340* and *LIC12339*, respectively). *LIMLP_11660* inactivation led to complete loss of virulence, which complementation restored (7). These two VM proteins homologs were associated with epithelial cell tight junction disruption increased calcium balance (7). These independent studies validated our hypothesis and previous findings, underscoring that VMPs are secreted protein exotoxins.”

<https://doi.org/10.1101/2025.04.04.647190>

- Reviewer 2 noted that “The study was continuous work from previous publications.” We disagree with the sense of this comment, and would emphasize that this study, while based on previous work, demonstrates true novelty by being the first to detect a bacterial pathogenic toxin in the blood with the potential both to lead to a diagnostic test as well as to provide some degree of insight into bacterial pathogenesis. As Isaac Newton famously once said, “If I have seen further, it is by standing on the shoulders of giants.” The present work, as with the vast bulk of published literature, does not purport to be the definitive work on the subject, but builds on previous work and simply reports a singular (and very important) finding.

2. Do other Spirochetes species harbor VM toxin homo or paralogs? If so, is there cross-reactivity of mAbs developed in this study against other Sprichaetes?

➤ Thank you for the comment. We have included the below statement in the discussion section and revised the manuscript accordingly.

➤ Line 511- 518

“Within spirochetes, the PF07598 gene family encoding VM proteins is unique to *Leptospira*, based on a comprehensive GenBank analysis. Interestingly, homologs of this gene family are present in unrelated α -proteobacteria, including *Bartonella bacilliformis*, *Bartonella ancashii*, and *B. australis*, each with multiple paralogs. Additionally, single gene copies are found in a few ϵ -proteobacteria species such as *Helicobacter hepaticus*, *H. mustelae*, and *H. cetorum* (8). Although *B. bacilliformis* and *B. ancashii* infect humans, they do not overlap clinically or epidemiologically with *Leptospira*. Cross-reactivity with these proteobacteria presents a potential avenue for future investigation.”

3. Please provide evidence that tests in the manuscript actually studied extracellularly-secreted VM exotoxins, and whether VM exotoxin does or does not exist intracellularly. Would dead or lysed bacterial cells interfere with the tests? Do the antibodies interact with cell pellets? In hamster assays (Figure 6), were extracellular, intracellular and from lysed/dead cells VM exotoxins differentiated?

➤ Thank you for this insightful suggestion. Together with the Picardeau group, we have already published (and referenced extensively) such compelling evidence that VMPs are extracellularly secreted VM exotoxins. Relevant references from the Picardeau group have been included in the revised manuscript.

➤ Based on our findings strongly indicate that VMPs function as secreted exotoxins, and all experiments were conducted using cell-free lysates.

- In the hamster model, we employed blood and urine samples to detect circulating VMPs using our capture ELISA assay, further supporting their extracellular secretion.

4. Not quite understand the selection process of these four antibody isotypes. For instance, four antibody isotypes were determined to have high affinity against the antigen, but only three combinations were selected in Capture-ELISA. Second, why 5E10 was not evaluated? Especially 6A5/5E10 pairing showed quite high affinity in Table 2. Please rephrase the selection process and, if needed, reorganize the figures and result sections.

- Thank you for the valuable suggestion. We have incorporated the below statement in the results section.

- Line 342 – 344

“Clone 5E10 did not yield sufficient mAb for scale-up. Therefore, we selected clones 5G10, 5F8, and 6A5 for large-scale production, as they demonstrated good yield, strong binding affinity, and consistent reproducibility.”

5. In Figure 6, What is the meaning of "H1~H5", total 5 hamsters? On Day 4 with 2×10^8 cells/mL infection dose, VM toxin levels in serum and urine were not proportional for H2, H3, and H4. What is the mechanism behind this?

- Thank you for pointing this out. H1-H5 indicates each hamster separately in the group. We have revised Figure 6 by incorporating a representative pathology of one hamster infected with 2×10^8 / mL on day 4. And also provided the pathology of an individual Hamster (H1-H4, from day 0 to day 5) as a Supplementary Figure 2 and legend.

- Line 426 – 431

“Hamsters infected with *L. interrogans* serovar Copenhageni strain Fiocruz L1-130 showed visible illness and gross pathological changes in major organs compared to controls. By day 4 post-infection, infected animals showed clear signs of illness, including jaundice, pulmonary hemorrhage, enlarged and congested liver and spleen, swollen kidneys, and pale

abdominal cavities (Fig. 6A). These findings indicate the presence of a systemic infection causing widespread organ involvement.”

- Panel A, in Fig 6, has been added. See below

- Line: 440 - 446

The discrepancy in VMP levels between serum and urine for H2, H3, and H4 at different time points, despite the same infection dose, may be attributed to individual host-specific differences in immune response, toxin clearance rates, or renal filtration efficiency (the hamsters were outbred). It's also possible that local tissue distribution or degradation dynamics of the toxin influence its detectability in different biofluids. Further studies involving host response profiling would help clarify the underlying mechanisms (Supplementary Fig. 2).

- These comments have been incorporated into the Results section of the revised manuscript along with Supplementary Figure 2 and the Legend (see below).

Raw images

➤ Line 893 - 900

“Supplementary Figure 2. Clinical progression in hamsters following *L. interrogans* serovar Copenhageni strain Fiocruz L1-130 infection at the time points. Hamsters were observed at specific intervals post-infection to evaluate the development of clinical signs linked to leptospirosis. Clinical symptoms worsened over time, and gross pathological changes became evident in major organs such as the kidneys, liver, and lungs, consistent with interstitial nephritis, hepatic inflammation, and pulmonary hemorrhage. The lower panel presents enlarged, representative images of the lungs and kidneys from a single hamster in each experimental group. Notably, hamster H4 succumbed to infection on day 5 before pathological assessment.”

6. SDS-PAGE results in Figure 3 were in poor quality, and even bands in markers were not straight or clear. Publication needs better quality.

➤ Thank you for this insightful suggestion. We have improved the figure resolution and applied consistent labeling to the markers. Additionally, we have included the Ponceau staining as a Supplementary Figure 1, which also shows the purity of our rVMPs. Arrows in the figure and the molecular weight and amino acid position of each recombinant VMP have been added to Figure 3, the Legend.

➤ Line 792 - 797

“(A) Soluble recombinant VM proteins purified using AKTA—including LA3490 (123 kDa; residues 19–639 aa, with mCherry), LA1402 (123 kDa; residues 28–641 aa, with mCherry), LA1400 (70 kDa; residues 1–573 aa, without mCherry), LA0591 (48 kDa; residues 23–313 aa, without mCherry), and RBL1 (52 kDa; residues 40–147 aa, with mCherry)—were analyzed by 4–12% SDS-PAGE (6, 9, 10).”

➤ Line 377-379

“The observation that 6A5 recognizes a single band of LA0591, while other mAbs detect multiple bands, likely reflects differences in epitope recognition and binding specificity (Fig. 3A, Supplementary Fig. 1, see below).”

➤ Line 887 – 891

“**Supplementary Figure 1. Purity and immunoreactivity of rVM proteins.** Western blot analysis of recombinant VM proteins (rLA0591, rLA1400, rLA1402, and rLA3490) probed with monoclonal antibodies 6A5 (A) and 5F8 (B). Recombinant VMPs were separated by SDS-PAGE, transferred to membranes, and detected using 6A5 and 5F8 antibodies to confirm specificity and reactivity. Arrows show the rVM proteins.”

7. Please review the manuscript text, correct typos and clarify/improve phrasing. For instance, in Table 1, correct terms in right upper and lower case forms. Secondly, what is the unit in Table 2? The high, the better? Suggest to add necessary description in Methods or table caption.

- Thank you for this insightful suggestion. We have revised the text, corrected typographical errors, and refined phrasing throughout the manuscript where necessary.
- We revised Tables 1 and 2 as suggested.
- Table 2 shows: Binding Response Unit (nm) – Nanometers
 - Measures the wavelength shift due to biomolecular binding (reflecting mass changes on the biosensor surface).
- Table 2 description has been revised by incorporating the above statement in the manuscript at the respective place.
- Line 880-884

Renamed both tables as well.

“**Table 1.** Binding kinetics of monoclonal antibodies against target antigen, recombinant leptospiral Virulence Modifying Protein, LA0591”

“**Table 2.** Biolayer interferometry (Octet) analysis of monoclonal antibody pairing using as binding antigen recombinant leptospiral Virulence Modifying Protein, LA0591”

Reviewer #3 (Comments for the Author):

Comments on the manuscript titled "Development of Leptospiral Virulence-Modifying Protein Detection Assay: Implications for Pathogenesis and Diagnostic Test Development"

In this study, the authors developed a monoclonal antibody (mAb)-based capture immunoassay to detect VM exotoxins in experimental hamster models, thereby validating the hypothesis that VM proteins function as secretory exotoxins. Monoclonal antibodies were generated against a natural variant LA0591, a VM protein containing a conserved C-terminal DNase toxin domain. Using mAbs 6A5 and 5F8, a capture ELISA successfully detected circulating VM exotoxins in serum and urine from infected hamsters, confirming the secretion of these proteins during infection.

The comments on the role of VM toxins in diagnosis and pathogenesis are discussed separately as follows:

I. Diagnosis

1. Line 321-323: "In the indirect ELISA, two-fold serial dilutions of 5F8, 6A5, and 5G10 mAbs were tested against a fixed amount of antigen, resulting in detection limits of 1.98 ng/mL for 5F8, 8.56 ng/mL for 6A5, and 0.33 µg/mL for 5G10 (Fig. 2A)."

Since a fixed amount of antigen was used, it is unclear how this experiment was designed to determine and calculate the LoD. Please clarify the methodology used for LoD estimation.

➤ We would point out, and have now revised the manuscript

➤ Line 443 - 449

“Here we demonstrate that *Leptospira*-secreted protein exotoxins, members of the PF07598 gene family-encoded Virulence Modify Proteins (VM proteins) can be detected in the blood and urine of experimentally-infected hamsters. This is the first time the authors are aware of that a secreted protein exotoxin in a systemic bacterial infection that putatively mediates clinical pathogenesis can be detected in blood and urine as the basis of a diagnostic test. We developed a monoclonal antibody-based immunoassay for the early detection of circulating VM proteins in blood and urine samples from hamsters experimentally infected with pathogenic *Leptospira*.”

➤ Thank you for your comment. We have clarified this methodology in the revised manuscript.

➤ Line 364 - 369

“The limit of detection (LoD) in the indirect ELISA was estimated based on the lowest concentration of antibody that produced a signal significantly above the background (mean of blank wells + 3 standard deviations). Although a fixed antigen concentration was used, serial dilutions of the antibodies allowed us to assess the sensitivity of each mAb in detecting the immobilized antigen. This approach reflects the minimum concentration of antibody required to generate a detectable signal under the given assay conditions.”

2. Line 326-329: "The two selected pairs, 6A5/5F8 and 5F8/6A5, were further evaluated using two-fold serial dilutions of LA0591 (starting at 1,000 ng/mL), with detection limits of 1.84 ng/mL for 6A5/5F8 and 1.31 ng/mL for 5F8/6A5 (Fig. 2C,D)." Based on the results shown in Fig. 2C and 2D, the LoD of the 6A5/5F8 pair would be expected to be better (i.e., lower) than that of 5F8/6A5. Please explain why the reported LoD of 6A5/5F8 was higher than that of 5F8/6A5.

➤ Thank you for pointing this out. We have clarified this point in the revised manuscript to avoid confusion.

➤ Line 361 - 367

“Although the overall binding signal for the 6A5/5F8 pair appeared higher in Fig. 2C compared to 5F8/6A5 in Fig. 2D, the calculated LoD was based on the lowest antigen concentration that consistently produced a signal exceeding the background (mean of blank + 3 SD). Minor variations in background noise and signal consistency across replicates may have influenced the calculated LoD values. Therefore, while 6A5/5F8 showed stronger signal intensity, the 5F8/6A5 pair demonstrated slightly better sensitivity under our assay conditions.”

3. The results of epitope mapping in Figure 5 should be visualized on the 3D structure shown in Figure 1. Highlighting the epitopes recognized by each mAb will provide clearer insight and help correlate with the epitope binning results presented in Table 2.

- Thank you for the valuable suggestion. We have now mapped the epitopes recognized by each mAb onto the 3D structure with color-coded highlights, corresponding to the results in Figure 4D and 4E, and Table 2.

Figure 4D and 4E

- A new surface representation has been added as Figures 4D and 4E, which we believe is the most appropriate placement for this data. Also incorporated the Legend and result.
- This integration offers a clearer understanding of the spatial distribution and potential overlap of antibody binding sites.
- Line 225 – 231: Methods

“Epitope mapping using PyMOL visualization

Three-dimensional structures of LA0591(Uniprot ID: Q8F8G6, Predicted AlphaFold: AF-Q8F8G6-F1) and full-length LA3490 (Uniprot ID: Q8F0K3, Predicted AlphaFold: AF-Q8F0K3-F1) were analyzed using PyMOL version (TM) 3.1.3.1, to localize conserved immunoreactive epitopes. Epitope 19 (NSHG**PLQGGGYF**), Epitope 20

(PLQGGGYFFNTA), and Epitope 67 (NRRGSSGGYPTSA) were color-coded in red, blue, and green, respectively, and mapped onto the surface structures of each protein.”

➤ Line 403 – 407: Results

“The 3D models of LA0591 and LA3490 highlight three conserved immune-reactive regions—Epitope 19 (red), Epitope 20 (blue), and Epitope 67 (green) (Fig. 4D and 4E). These regions are marked and shown on both front and rotated views of the protein surfaces, helping visualize where these epitopes are located and how accessible they are to the immune system.”

➤ Line 819 - 825

“The mapped epitopes were further localized on 3D structural models of LA0591 (**D**) and LA3490 (**E**), highlighting three conserved immunoreactive regions: Epitope 19 (red), Epitope 20 (blue), and Epitope 67 (green). LA0591 is shown in light blue, and LA3490 in pink, with epitopes displayed on both front and rotated surface views, emphasizing their spatial distribution and surface exposure of immunoreactive epitopes on the modeled structures of VM proteins. Underlined sequences indicate overlapping regions across the identified linear epitopes. The ELISA protocol illustration was created using BioRender.com.”

4. In Figure 3A, the expected molecular weights of paralogous VTMs should be mentioned to aid in interpreting the results. Including the SDS-PAGE images corresponding to the Western blots (in the Supplement) would help confirm that the absence of bands was not due to undetectable or low protein levels. Furthermore, it is unclear why 6A5 recognizes a single band of LA0591, while other mAbs detect multiple bands, despite using the same recombinant protein. Comparing the epitope sequences recognized by each mAb to those in other paralogous VTMs, as done in Figure 5, would help clarify differences in specificity and cross-reactivity—particularly why mAb 6A5 exhibited broader reactivity.

➤ Thank you for the valuable suggestion. We have now included the arrow in Figure 3A and the expected molecular weights of the paralogous VMPs in the respective Legend to aid interpretation.

➤ Line 790 - 795

“(A) Soluble recombinant VM proteins purified using AKTA—including LA3490 (123 kDa; residues 19–639 aa, with mCherry), LA1402 (123 kDa; residues 28–641 aa, with mCherry), LA1400 (70 kDa; residues 1–573 aa, without mCherry), LA0591 (48 kDa; residues 23–313 aa, without mCherry), and RBL1 (52 kDa; residues 40–147 aa, with mCherry)—were analyzed by 4–12% SDS-PAGE (6, 9, 10).”

➤ Additionally, the corresponding Ponceau stain and Western blot have been added to Supplementary Figure 1 to confirm that the absence of bands in the Western blots was not due to low or undetectable protein levels. These additions help support the validity of the Western blot results as well as the purity of proteins. Please refer above.

➤ Thank you for this insightful comment.

➤ Line 377 - 379

“The observation that 6A5 recognizes a single band of LA0591, while other mAbs detect multiple bands, likely reflects differences in epitope recognition and binding specificity (Fig. 3A, Supplementary Fig. 1).”

➤ We have now mapped the epitopes recognized by each mAb onto the 3D structure with color-coded highlights, corresponding to the results in Figure 4D and 4E. Please refer above.

5. According to Figure 5, the conserved epitopes recognized by mAbs 5F8/5G10 and 6A5 should allow detection of VTM in the Lai strain, especially under induced conditions (Figure 3B). Please clarify why little or no reactivity was observed. Additionally, since the mAbs detected targets under denaturing conditions in SDS-PAGE/Western blot, they are likely to recognize linear epitopes.

➤ Thank you for your observation.

➤ Line 389 - 394

“While the epitopes recognized by mAbs 5F8/5G10 and 6A5 are conserved in the Lai strain, the lack of reactivity observed may be due to low expression levels of the VMPs under the tested conditions, even after NaCl treatment (Fig. 3B). It is also possible that post-translational modifications or structural differences in the Lai strain impact the accessibility or stability of these epitopes in native conditions.”

- We agree that detection under denaturing conditions suggests recognition of linear epitopes, and we have clarified both points in the revised manuscript.

6. Although the mAbs demonstrated diagnostic potential in hamsters, the limitations of the study should be addressed. While the capture ELISA detected VM antigens in blood and urine samples from hamsters, it remains unclear whether the assay is sufficiently sensitive for use with human clinical samples. Can it achieve the necessary LoD to detect VM proteins in clinical settings? What is the expected concentration of VM antigens in human samples, and how does this correlate with leptospiral burden? These points should be discussed with reference to published data on leptospiral loads in human blood and urine. Furthermore, the specificity of the assay has not been assessed against antigens from other pathogens. Potential cross-reactivity should be evaluated to confirm the diagnostic specificity of the capture ELISA using the selected mAbs.

- Thank you for your thoughtful comments. We acknowledge the limitations of our current study and have now addressed them in the revised discussion. While our capture ELISA demonstrated promising sensitivity in detecting VM proteins in hamster blood and urine. At this stage, the limit of detection (LoD) of our assay falls within the picogram/mL range, which is encouraging.
- Further validation using human clinical samples is ongoing and shows promise for detecting VMPs in the nanogram range. However, detection sensitivity may vary with infection stage and severity. Published data report leptospiral loads in human blood and urine between 10^2 and 10^6 genome equivalents/mL. We are currently investigating how VM antigen levels correlate with these bacterial burdens and plan to present these findings in a subsequent publication.

- However, we have addressed these points with reference to published data on leptospiral loads in human blood and urine, highlighting the reported ranges and variability across studies to support our discussion as per the suggestion:

- Line 458 – 465
“Previous meta-analysis has provided quantitative estimates of *Leptospira* spp. loads in various hosts, including rats (5.7×10^6 /mL urine), mice (3.1×10^3 /mL), cattle (3.7×10^4 /mL), and humans (7.9×10^2 /mL) (11-13). However, differences in sample size, detection methods, and *Leptospira* species and serovars limit cross-species comparisons. The presence of *Leptospira* in the kidney often reflects levels in urine. Supporting this observation, Costa et. al observed a strong correlation between kidney and urinary leptospiral loads in brown rats using qPCR. In humans, reported urinary leptospiral loads vary widely, from $\sim 7.9 \times 10^2$ to 5.7×10^6 /mL, due to factors such as infection stage, host response, and detection methodology (11-13).”

- These points have been discussed and clarified in the revised manuscript with references to relevant literature.

- Thank you for highlighting this important aspect.

- Line 478 - 482
“In our ongoing efforts, we are broadening the specificity evaluation of the assay by testing the selected mAbs against antigens from other relevant pathogens, as well as samples from diverse geographical settings. This will allow us to assess potential cross-reactivity and further establish the diagnostic specificity of the capture ELISA.”

II. Pathogenesis

To support the conclusion that "The presence of leptospiral VM exotoxins in the serum and urine of infected hamsters supports the hypothesis that circulating VM exotoxins mediate disease pathogenesis," it is necessary to correlate VM toxin levels with disease severity in infected hamsters or humans.

➤ Thank you for the suggestion. Please refer to the response provided above.

1. In this study, capture ELISA was used to detect and quantify VM exotoxins in the serum and urine of hamsters (Fig. 6C, D) infected with different inoculum doses of *Leptospira*. However, VM protein levels should more accurately correlate with the actual number of *Leptospira* present in the blood or urine rather than with the inoculum dose alone, as toxin levels varied among hamsters within the same group. Furthermore, since a direct correlation between *Leptospira* load and VM protein levels was not assessed, it remains unclear whether the assay is sufficiently sensitive to detect VM exotoxins at lower inoculum doses.

➤ Thank you for this valuable observation.

➤ Line 489 – 496

“We agree that “VM protein levels are likely to correlate more closely with the actual *Leptospira* burden in blood or urine than with the initial inoculum dose. This hypothesis is supported by the observed variability in VM levels among hamsters within the same infection group. While the present study primarily aimed to establish the presence of VM proteins using a capture ELISA, future work will focus on quantifying *Leptospira* load in parallel with VM protein detection. This approach will enable a more accurate assessment of the assay’s sensitivity, particularly in low-dose infections, and provide a clearer understanding of the relationship between toxin production and bacterial burden during infection.”

2. Were VM exotoxin levels significantly different among hamsters infected with varying inoculum doses? In Figure 6C, serum toxin levels do not appear to be clearly dose-dependent. The study would be strengthened by correlating inoculum dose and VM levels with severity outcomes, such as survival and organ pathology, to demonstrate that higher VM levels are associated with more severe disease.

➤ Thank you for this insightful suggestion. While our data in Figure 6C show detectable VM exotoxin levels across different inoculum doses, the levels were not consistently dose-

dependent, which likely reflects host-specific responses and variable bacterial dissemination, given that we used outbred Syrian Golden hamsters. We agree that correlating VM protein levels with clinical outcomes such as survival rates and organ pathology would strengthen the subsequent study. As part of our ongoing work, we are integrating VM levels with disease severity metrics to better understand their relationship and further validate the role of VM exotoxins in pathogenesis.

➤ Line 110-111

3. Please explain why VM toxin levels were measured only on Days 0-5 in the high-dose group (2×10^8 cells/mL), as shown in Figure 6D. During the early stages of infection, *Leptospira* are expected to be more abundant in the blood than in the urine. The authors did not provide data on survival or disease severity in the infected hamsters. In lower-dose infections, hamsters may survive longer and progress into the leptospiruric phase, during which urinary shedding of VM proteins could be evaluated. At early time points, VM toxin levels may be too low for detection by capture ELISA, and their presence in urine following high-dose infection may reflect spillover from bacteremia or bacterial lysis rather than active secretion of VM toxins by *Leptospira* in the kidneys. Renal tissues should be analyzed for VM protein and correlated with histopathological findings.

➤ Thank you for this detailed and insightful comment. We performed this experiment with different dose and one single end time point for the serum and VM toxin levels in urine were measured on Days 0–5 in the high-dose group (2×10^8 cells/mL) to focus on the early bacteremia phase of infection and potentially leptospiruric phase at higher dose, when systemic dissemination and toxin circulation are expected to peak.

➤ We agree that during early infection, *Leptospira* are more abundant in the blood than in urine, and that urinary detection of VM proteins at this stage may reflect spillover from bacteremia or bacterial lysis rather than renal secretion.

- We appreciate the suggestion regarding lower-dose infections. We have incorporated these suggestions into the discussion section, as well as in the study's limitations and future perspectives

- (Line 553-564)

“We do not have information on VM protein kinetics in blood or urine at earlier time points after the challenge infection. It is possible that VM protein levels may remain below the detection threshold of our capture ELISA, and theoretically, the presence of VM protein levels in urine following high-dose infection may result from spillover due to bacteremia or bacterial lysis rather than active secretion by *Leptospira* in the renal environment. Our ongoing studies are tracking VM protein kinetics in blood and urine throughout infection, particularly during the later stages. These investigations will include analysis of kidney tissue to assess local VM expression and its association with tissue pathology and abnormalities of urine and blood kidney function tests. Such studies will help distinguish between passive leakage and active secretion of VM proteins, ultimately refining their potential as diagnostic markers for both acute and chronic leptospirosis. We plan to test and validate the diagnostic utility of VM protein antigen detection in blood and urine in diverse settings of human leptospirosis.”

4. In light of the above points, the detection of VM toxins in the blood and urine of infected hamsters-based on capture ELISA-and the absence of detection in the lung may be overstated in the summary presented in Figure 7. Please revise the interpretation accordingly.

- Thank you for the suggestion. We agree that the interpretation in Figure 7 could be more accurately aligned with the experimental findings. We have revised the summary. Figure 7 is a hypothetical model intended to summarize our current understanding based on available data. We acknowledge that lung tissues have not been analyzed at this stage, and we have updated the figure legend to reflect this, ensuring there is no confusion or overstatement in the interpretation.

➤ Line 515–522

“At the cellular level, VMPs may directly or indirectly target critical organs such as the lungs, liver, and kidneys. In the lungs, VMPs could disrupt tight junction proteins in endothelial cells, potentially leading to pulmonary hemorrhage. In the kidneys, leptospire may colonize and form biofilms, with secretion of VMPs exotoxins enabling proximal tubule colonization and potentially contributing to acute kidney injury. Similarly, in the liver, VMP-mediated effects may contribute to hepatic dysfunction (Fig. 7). Future work will focus on the identification specific target cells and receptor-ligand interactions involved in VM protein binding, which is fundamental to understanding the mechanistic basis of the clinical pathogenesis of leptospirosis.”

➤ Line 856 – 866

“The underlying hypothesis is that systemically circulating leptospiral VM proteins are mediate/contribute to multiple organ dysfunction, particularly liver (jaundice), kidney (acute kidney injury), and lung (hemorrhage). Liver dysfunction, where they damage sinusoidal endothelial cells, disrupting bile flow and causing bilirubin buildup — a key factor behind jaundice. In the kidneys, *Leptospira* organisms colonize the proximal renal tubules, form biofilms, and release VMPs that compromise endothelial integrity, ultimately contributing to acute kidney injury. Within the lungs, infection leads to capillary damage driven by VMP exotoxins, resulting in blood leakage into the alveolar spaces and subsequent pulmonary hemorrhage. Clinically, patients affected by leptospirosis often present with a combination of jaundice, renal impairment, and pulmonary complications, all of which stem from the systemic circulation and either primary or secondary action of VMP exotoxins.”

Reviewer #4 (Comments for the Author):

The study by Chaurasia et al. adds a new and important mechanism to the pathogenesis of leptospirosis by providing the first evidence of secreted leptospiral exotoxins in the bloodstream of infected animals, and establishes a basis for the development of novel diagnostic and therapeutic approaches for the disease. The authors concluded that further work is needed to investigate the tissue-specific expression of VM proteins. Although the authors have not demonstrated the presence of MV proteins in the tissues, I would like to suggest that they describe the clinical condition of the hamsters throughout the course of the disease and at the time of euthanasia in the results, and if they could include graphs showing the kinetics of MV proteins in the blood and urine throughout the experiment, associated with the clinical condition of the animals during the experiment. I suggest that the authors describe the macroscopic appearance of the lungs, liver, kidneys and spleen at the time of euthanasia, with the possibility of providing a figure showing the appearance of the organs when the thoracic and abdominal cavities are opened. Is there any data on leukocyte count, urea, creatinine or bilirubin during the experiments.

- Thank you for recognizing the important value of this work.

- Thank you for this thoughtful and valuable suggestion. Our current study focused primarily on detecting and quantifying VM proteins in blood and urine. We agree that correlating these findings with clinical signs and organ pathology would strengthen the disease model.

- We have incorporated systematic macroscopic evaluations of key organs—lungs, liver, kidneys, and spleen—at necropsy and will consider including representative images of thoracic and abdominal cavities (Figure 4A and Supplementary Figure 2). Please refer above.

- Line 553 – 564
“We do not have information on VM protein kinetics in blood or urine at earlier time points after the challenge infection. It is possible that VM protein levels may remain below the detection threshold of our capture ELISA, and theoretically, the presence of VM protein levels in urine following high-dose infection may result from spillover due to bacteremia or bacterial lysis rather than active secretion by *Leptospira* in the renal environment. Our

ongoing studies are tracking VM protein kinetics in blood and urine throughout infection, particularly during the later stages. These investigations will include analysis of kidney tissue to assess local VM expression and its association with tissue pathology and abnormalities of urine and blood kidney function tests. Such studies will help distinguish between passive leakage and active secretion of VM proteins, ultimately refining their potential as diagnostic markers for both acute and chronic leptospirosis. We plan to test and validate the diagnostic utility of VM protein antigen detection in blood and urine in diverse settings of human leptospirosis.”

- We appreciate your suggestions. We have incorporated these points into the discussion and future perspectives section of the revised manuscript.
- The references below have been incorporated into the revised manuscript to address the reviewer’s comments.

References

- 1. Chaurasia R, Marroquin AS, Vinetz JM, Matthias MA. 2022. Pathogenic *Leptospira* evolved a unique gene family comprised of ricin B-like lectin domain-containing Cytotoxins. *Frontiers in Microbiology* 13:859680.
- 2. Lehmann JS, Fouts DE, Haft DH, Cannella AP, Ricaldi JN, Brinkac L, Harkins D, Durkin S, Sanka R, Sutton G, Moreno A, Vinetz JM, Matthias MA. 2013. Pathogenomic inference of virulence-associated genes in *Leptospira interrogans*. *PLoS Negl Trop Dis* 7:e2468.
- 3. Giraud-Gatineau A, Nieves C, Harrison LB, Benaroudj N, Veyrier FJ, Picardeau M. 2024. Evolutionary insights into the emergence of virulent *Leptospira* spirochetes. *bioRxiv* doi:10.1101/2024.04.02.587687.
- 4. Fouts DE, Matthias MA, Adhikarla H, Adler B, Amorim-Santos L, Berg DE, Bulach D, Buschiazio A, Chang YF, Galloway RL, Haake DA, Haft DH, Hartskeerl R, Ko AI, Levett PN, Matsunaga J, Mechaly AE, Monk JM, Nascimento AL, Nelson KE, Palsson B, Peacock SJ, Picardeau M, Ricaldi JN, Thaipandungpanit J, Wunder EA, Jr., Yang XF, Zhang JJ, Vinetz JM. 2016. What Makes a Bacterial Species Pathogenic?: Comparative Genomic Analysis of the Genus *Leptospira*. *PLoS Negl Trop Dis* 10:e0004403.
- 5. Chaurasia R, Vinetz JM. 2022. In silico prediction of molecular mechanisms of toxicity mediated by the leptospiral PF07598 gene family-encoded virulence-modifying proteins. *Front Mol Biosci* 9:1092197.
- 6. Chaurasia R, Marroquin AS, Vinetz JM, Matthias MA. 2022. Pathogenic *Leptospira* Evolved a Unique Gene Family Comprised of Ricin B-Like Lectin Domain-Containing Cytotoxins. *Front Microbiol* 13:859680.
- 7. Giraud-Gatineau Alexandre HG, Monot Marc, Picardeau Mathieu, Benaroudj Nadia. 2025. In Vivo Dual RNA-Seq uncovers key toxin-like effectors of epithelial barrier disruption and tissue colonization by an extracellular bacterial pathogen. doi:<https://doi.org/10.1101/2025.04.04.647190>.
- 8. Lehmann JS, Matthias MA, Vinetz JM, Fouts DE. 2014. Leptospiral pathogenomics. *Pathogens* 3:280-308.
- 9. Chaurasia R, Liang C, How K, Vieira DS, Vinetz JM. 2023. Production and Purification of Cysteine-Rich Leptospiral Virulence-Modifying Proteins with or Without mCherry Fusion. *Protein J* 42:792-801.
- 10. Chaurasia R, Salovey A, Guo X, Desir G, Vinetz JM. 2022. Vaccination With *Leptospira interrogans* PF07598 Gene Family-Encoded Virulence Modifying Proteins Protects Mice From Severe Leptospirosis and Reduces Bacterial Load in the Liver and Kidney. *Front Cell Infect Microbiol* 12:926994.
- 11. Barragan V, Nieto N, Keim P, Pearson T. 2017. Meta-analysis to estimate the load of *Leptospira* excreted in urine: beyond rats as important sources of transmission in low-income rural communities. *BMC Res Notes* 10:71.
- 12. Moinet M, Wilkinson DA, Aberdein D, Russell JC, Vallee E, Collins-Emerson JM, Heuer C, Benschop J. 2021. Of Mice, Cattle, and Men: A Review of the Eco-Epidemiology of *Leptospira borgpetersenii* Serovar Ballum. *Trop Med Infect Dis* 6.

- 13. Costa F, Wunder EA, Jr., De Oliveira D, Bisht V, Rodrigues G, Reis MG, Ko AI, Begon M, Childs JE. 2015. Patterns in *Leptospira* Shedding in Norway Rats (*Rattus norvegicus*) from Brazilian Slum Communities at High Risk of Disease Transmission. *PLoS Negl Trop Dis* 9:e0003819.
- Giraud-Gatineau A, Georges H, Monot M, Benaroudj N, Picardeau M. *In Vivo* Dual RNA-Seq uncovers key toxin-like effectors of epithelial barrier disruption and tissue colonization by an extracellular bacterial pathogen, 2025; <https://doi.org/10.1101/2025.04.04.647190>

Re: Spectrum00018-25R1 (Development of Leptospiral Virulence-Modifying Protein Detection Assay: Implications for Pathogenesis and Diagnostic Test Development)

Dear Prof. Joseph M. Vinetz:

Thank you for the privilege of reviewing your work. Below you will find my comments, instructions from the Spectrum editorial office, and the reviewer comments.

Regarding Figure 3, please do not make further modifications to the figure itself, but ensure that molecular weights are indicated next to the protein bands rather than directly on them; moreover, please use the figure as presented in the earlier version.

Revision Guidelines

Sincerely,
Denis Sereno
Editor
Microbiology Spectrum

Reviewer #2 (Comments for the Author):

In Figure 3, authors used blue lines to mark protein markers, with quite a few bands that were not visible in previous paper edition, and manually marked by the author in this edition. Myself would not recommend this action. I recommend repeating this experiment and replace with images of better quality.

Reviewer #4 (Comments for the Author):

I recommend to "Accept" the manuscript. Congratulations to all the authors for this important work.

Reviewer #5 (Comments for the Author):

This manuscript describes the characterization and evaluation of several monoclonal antibodies against a virulence-related protein of pathogenic *Leptospira*, i. e. LA0591 from the Virulence-Modifying (VM) protein family. When used in a capture-based assay, these antibodies allowed the detection of VM proteins in blood and urine of *Leptospira*-infected hamsters. VMs have been identified and characterized as exotoxin by the authors in previous studies but the exact mechanisms on how these proteins promote *Leptospira* pathogenicity is not fully understood.

Overall, this study is well-executed. The detection of VMs within a host supports a mechanism whereby these proteins are secreted by pathogenic *Leptospira* when infecting a mammalian host. Moreover, the use of anti-VMs antibodies represents a promising tool for leptospirosis diagnosis, should the authors validate such possibility in future studies.

In the "Response to Reviewer Comments", the authors stated that "...this is the first time in the leptospirosis field-indeed in any field of bacterial infections-in which a secreted, pathogenetic protein exotoxin can be detected in the blood of an infected animal." and "...being the first to detect a bacterial-secreted protein exotoxin in the blood of an infected animal...". However, the authors should be aware that toxins of other bacterial pathogens have been already detected in infected animals, including anthrax (PMID: 16760326) and listeriolysin (PMID: 7779970). Other than that, the authors have complied with all the requirements of the other reviewers in this revised version. I have just identified the following points that need to be corrected or modified for the clarity of the manuscript.

1. Line 83 "...and a C-terminal toxin domain with DNase activity". This should be rephrased as it is presently unclear whether all VMs have a DNase activity. Indeed, such activity has only been experimentally demonstrated for a subset of recombinant VM proteins (LA3490, LA0591, LA0620, LA1400, LA1402). Moreover, the mechanisms of this activity and the amino acid residues involved are not known. Putative important amino acid residues have been proposed by the authors in previous studies but some of these residues are not conserved in all VM-encoding ORFs. Moreover, the involvement of these residues in the DNase activity have never been demonstrated.
2. Line 138-143. The authors should give more experimental details on how they have harvested urine samples from hamsters. Why the urine samples needed to be centrifuged and pellets conserved? Which fraction of the urine was analyzed for the presence of VM, the supernatant or the pellet?
3. Line 202-204. The method for determination of the Kd dissociation constant is confusing as the authors mentioned Kd as "dissociation constant" (line 202) and "dissociation rate constant" (line 203). Do they mean off-rate constant koff by "dissociation rate constant"? Also, in Table 1, "dissociation rate constant" is named kdis. This should be clarified/corrected for consistency.
4. Lines 259-261. It is mentioned that *Leptospira* culture supernatants were also collected. Were they also analyzed for the presence of VMs? I could not find where this analysis was presented in the manuscript.
5. Line 338. "VM exotoxins are single-gene-encoded polypeptides that self-assemble into a fully active exotoxins...". What the authors mean by "self-assemble"? Do they mean that VMs form homo-oligomers? Has this been demonstrated? If not, this should be rephrased.
6. Lines 340-342. "The PF07598 gene family encodes a variant that lacks the carbohydrate-binding receptors (RBLs) and includes only the C-terminal toxin domain (313 amino acids)". It should be mentioned in this sentence that LA0591 is the VM variant naturally lacking the C-terminal domain.
7. Line 369. The abbreviation "LoD" should be explained in this paragraph and not in the following paragraph (Lines 374-379).
8. Lines 381-403. The authors should explain that the treatment with 120 mM NaCl aims at mimicking host osmolarity.
9. Lines 400-401. "The diverse profiling of VMPs across all pathogenic serovars highlights their varied roles in pathogenesis." This statement is only speculative and it should be removed as it is not possible to determine exactly which VM is detected in total lysates of *Leptospira* by the antibody used.
10. Lines 401-403. "Upregulation of VM proteins in human-infecting lethal *Leptospira interrogans* serovar Copenhageni strain L1-130 implies a potential role in severe human leptospirosis and their use as therapeutic targets". This sentence should be moved to the discussion.
11. Line 412. If "LA0591" is a protein variant in which amino acids were mutated, it should be named otherwise as "LA0591" would normally refer to a deletion mutant strain (according to the genetic conventional nomenclature). Also, the mutated AAs in this variant should be listed in Material and Methods or Figure legends.
12. Figure 6D. Can the authors provide the same analysis in non-infected hamster as a negative control?
13. Lines 455-456. "This is the first time the authors are aware of that a secreted protein exotoxin in a systemic bacterial infection that putatively mediates clinical pathogenesis can be detected in blood and urine as the basis of a diagnostic test.". The authors do not demonstrate the use of VM detection as diagnosis tool. This should be rephrased or removed.
14. Line 458. Detecting VM at 4-day post infection in blood is not considered an early detection in the hamster model as the animals are exhibiting morbidity sign at this stage of infection. This should be rephrased.
15. Lines 507-514 and 515-522. The two paragraphs are repetition.

16. Lines 817-818. The arrows in Figure 4B probably correspond to the LA0591 (48 kDa). Have the other any evidence that the band around 48 kDa correspond to other truncated VMs that could crossreact with the antibody? This should be clarified or corrected.
17. Line 853. "2x10⁸ cells/mL" should be corrected to "2x10⁸ leptospires/mL"
18. Lines 850, 859, "Copenhagni" should be corrected to "Copenhagenii"
19. Line 861. "2x10⁸ cells/mL" should be corrected to "2x10⁸ leptospires/mL".
20. What is represented in Figure 6C? It is unclear whether the linear regression is made with recombinant LA0591 of Figure 6B? This should be described in the legend.
21. Line 864. Which patient samples the authors are referring to here?
22. Line 866. "VMP proteins" should be corrected into "VM proteins"
23. Lines 875-877. "Clinically, patients affected by leptospirosis often present with a combination of jaundice, renal impairment, and pulmonary complications, all of which stem from the systemic circulation and either primary or secondary action of VMP exotoxins". This statement should be rephrased as there is no experimental evidence that VMs are in fact responsible for all the leptospirosis-associated tissue damage upon acute leptospirosis.

July, 2025

To,
Dr. Denis Sereno
Editor, Microbiology Spectrum

Dear Dr. Sereno,

Thank you for the opportunity to revise our manuscript. We have carefully addressed the reviewers' comments and revised the manuscript accordingly. We appreciate your consideration and look forward to your feedback at your earliest convenience.

Regarding Figure 3, please do not make further modifications to the figure itself, but ensure that molecular weights are indicated next to the protein bands rather than directly on them; moreover, please use the figure as presented in the earlier version (**Reviewer #2, Comments for the Author**).

→ Thank you for the clarification. We will retain the earlier version of Figure 3 as requested and ensure that molecular weights are indicated next to the protein bands, not directly on them.

Reviewer #4 (Comments for the Author)

→ Thank you very much for your kind recommendation and support. We sincerely appreciate your time and thoughtful review.

Reviewer #5 (Comments for the Author)

“this is the first time in the leptospirosis field-indeed in any field of bacterial infections-in which a secreted, pathogenetic protein exotoxin can be detected in the blood of an infected animal.” and “...being the first to detect a bacterial-secreted protein exotoxin in the blood of an infected animal...”. However, the authors should be aware that toxins of other bacterial pathogens have already been detected in infected animals, including anthrax (PMID: 16760326) and listeriolysin (PMID: 7779970).

→ Thank you for this important clarification. We now clarify that, to our knowledge, this is the first report of such detection in leptospirosis.

Lines 486-492

“To our knowledge, this is the first report of a secreted protein exotoxin being detected in both the blood and urine during systemic *Leptospira* infection, highlighting its potential involvement in disease pathogenesis. While secreted toxins from other bacterial pathogens, such as anthrax and listeriolysin, have previously been detected in infected animals (PMID: 16760326; PMID: 7779970), this represents the first such observation in leptospirosis. This finding expands our understanding of leptospiral virulence mechanisms and offers new insight into host–pathogen interactions in this neglected tropical disease.”

1. Line 83 “...and a C-terminal toxin domain with DNase activity”. This should be rephrased as it is presently unclear whether all VMs have a DNase activity. Indeed, such activity has only been experimentally demonstrated for a subset of recombinant VM proteins (LA3490, LA0591, LA0620, LA1400, LA1402). Moreover, the mechanisms of this activity and the amino acid residues involved are not known. Putative important amino acid residues have been proposed by the authors in previous

studies but some of these residues are not conserved in all VM-encoding ORFs. Moreover, the involvement of these residues in the DNase activity have never been demonstrated.

→ Thank you for this important insight. We agree that DNase activity has been experimentally confirmed only for a subset of recombinant VM proteins, including LA3490, LA0591, LA0620, LA1400, and LA1402. We also acknowledge that the specific mechanisms underlying this activity and the critical amino acid residues involved remain unclear. We have revised the manuscript to clarify these points.

Lines 84-88

“DNase activity has been experimentally demonstrated for a limited subset of recombinant VM proteins (LA3490, LA0591, LA0620, LA1400, and LA1402). However, the molecular mechanisms driving this activity, as well as the specific amino acid residues responsible, remain to be elucidated. Although putative critical residues have been suggested in earlier studies, their involvement in DNase activity has yet to be experimentally confirmed.”

2. Line 138-143. The authors should give more experimental details on how they have harvested urine samples from hamsters. Why the urine samples needed to be centrifuged and pellets conserved? Which fraction of the urine was analyzed for the presence of VM, the supernatant or the pellet:

→ We appreciate the reviewer’s comment and have added further experimental details in the revised manuscript.

Lines 146-154

“Urine samples were collected from hamsters at the time of euthanasia by directly aspirating urine from the bladder through an open abdominal cavity to avoid contamination. Samples were then centrifuged at 28000xg for 30 minutes to separate cellular debris and particulate material. Both the supernatant and pellet fractions were stored for downstream analysis. For the detection of VM, we primarily analyzed the pellet fraction, which was resuspended in 100 μ L of 1xPBS, as preliminary experiments indicated that VM was predominantly associated with particulate components, possibly due to secretion in vesicles or association with host/bacterial cells. Both the blood and urine were aliquoted to avoid freeze-thaw cycles and stored at -80°C until further use.

3. Line 202-204. The method for determination of the K_d dissociation constant is confusing as the authors mentioned K_d as "dissociation constant" (line 202) and "dissociation rate constant" (line 203). Do they mean off-rate constant k_{off} by "dissociation rate constant"? Also, in Table 1, "dissociation rate constant" is named k_{dis} . This should be clarified/corrected for consistency.

→ We thank the reviewer for this valuable observation. We have clarified the terminology throughout the manuscript for consistency.

Lines 216-225

The equilibrium dissociation constant (K_d , units: M) quantifies the binding affinity between the antibody and antigen. The dissociation rate constant (k_d units: s^{-1}) describes the rate at which the antibody-antigen complex dissociates, while the association rate constant (K_a units: $1/M \cdot s$), represents the rate at which the antibody binds to the antigen.

4. Lines 259-261. It is mentioned that *Leptospira* culture supernatants were also collected. Were they also analyzed for the presence of VMs? I could not find where this analysis was presented in the manuscript.
→ Thank you for bringing this to our attention. We have corrected the error by removing the statement.
5. Line 338. "VM exotoxins are single-gene-encoded polypeptides that self-assemble into a fully active exotoxins...". What the authors mean by "self-assemble"? Do they mean that VMs form homo-oligomers? Has this been demonstrated? If not, this should be rephrased.
→ Thank you for your insightful question. By "self-assemble," we intended to convey that VM exotoxins function as single-gene-encoded polypeptides that may oligomerize to form their active conformation. However, we acknowledge that direct evidence for homo-oligomer formation by VMs has not yet been demonstrated. We have revised the manuscript to clarify this point and avoid any overinterpretation.

Lines 357-361

"VM exotoxins are encoded by single genes as polypeptides that are potentially oligomerized into fully active exotoxins (~640 amino acids) (20, 21). Oligomerization is a well-documented mechanism of activation for various bacterial exotoxins, including anthrax toxin (PMID: 16760326) and listeriolysin O (PMID: 7779970). However, direct evidence for homo-oligomer formation by VM exotoxins remains to be established and warrants further investigation." Also added a few references.

6. Lines 340-342. "The PF07598 gene family encodes a variant that lacks the carbohydrate-binding receptors (RBLs) and includes only the C-terminal toxin domain (313 amino acids)". It should be mentioned in this sentence that LA0591 is the VM variant naturally lacking the C-terminal domain.

→ Thank you – rephrase the sentence as below:

Lines 362-366

"The PF07598 gene family typically encodes variants possessing both carbohydrate-binding receptor (RBL) domains and a C-terminal toxin domain. In contrast, LA0591 is a unique natural variant that lacks the RBL domains entirely and encodes only the C-terminal toxin domain, distinguishing it from other family members and suggesting potentially distinct functional roles."

7. Line 369. The abbreviation "LoD" should be explained in this paragraph and not in the following paragraph (Lines 374-379).

Thank you – correction made as suggested. **Lines 395-401**

8. Lines 381-403. The authors should explain that the treatment with 120 mM NaCl aims at mimicking host osmolarity.

→ Thank you – add the following statements:

Lines 407-412

"Exposure to 120 mM NaCl in EMJH medium was employed to simulate the physiological osmolarity conditions encountered by *Leptospira* during host infection. The transition from low-salt environmental conditions to the higher osmolarity of mammalian tissues is known to influence the expression of virulence-associated genes and host-adaptive responses. Supplementation of EMJH medium with NaCl thus provides a relevant in vitro model for studying *Leptospira* under host-mimicking conditions (PMID: 15618142)."

9. Lines 400-401. "The diverse profiling of VMPs across all pathogenic serovars highlights their varied roles in pathogenesis." This statement is only speculative and it should be removed as it is not possible to determine exactly which VM is detected in total lysates of *Leptospira* by the antibody used.

→ Thank you – removed the statement as you suggested.

10. Lines 401-403. "Upregulation of VM proteins in human-infecting lethal *Leptospira interrogans* serovar Copenhageni strain L1-130 implies a potential role in severe human leptospirosis and their use as therapeutic targets". This sentence should be moved to the discussion.

Thank you – Moved to discussion. **Lines 501-03**

11. Line 412. If " Δ LA0591" is a protein variant in which amino acids were mutated, it should be named otherwise as " Δ LA0591" would normally refer to a deletion mutant strain (according to the genetic conventional nomenclature). Also, the mutated AAs in this variant should be listed in Material and Methods or Figure legends.

→ Thanks for the valuable suggestion. We incorporated the following statement into the Methods.

Lines 163-169

"To generate the LA0591 mutant, the following site-directed mutations were introduced into the protein sequence: at position 203, arginine (R) was replaced with lysine (K); at position 205, histidine (H) was substituted with alanine (A); at position 221, threonine (T) was replaced with alanine (A); at position 222, arginine (R) was replaced with lysine (K); and at position 254, arginine (R) was substituted with lysine (K). These amino acid substitutions were designed to assess the potential structural and functional effects of altering charge and polarity at these specific residues."

12. Figure 6D. Can the authors provide the same analysis in non-infected hamster as a negative control?

→ Thank you for your suggestion. The same analysis was performed on samples from non-infected hamsters as negative controls. The values obtained from these controls were subtracted and accounted for in the representative figure to ensure accurate interpretation. **Lines 903-905**

13. Lines 455-456. "This is the first time the authors are aware of that a secreted protein exotoxin in a systemic bacterial infection that putatively mediates clinical pathogenesis can be detected in blood and urine as the basis of a diagnostic test.". The authors do not demonstrate the use of VM detection as a diagnostic tool. This should be rephrased or removed.

→ Thank you for your valuable suggestion. We rephrase the sentence as below:

Lines 486-497

"To our knowledge, this is the first report of a secreted protein exotoxin being detected in both the blood and urine during systemic *Leptospira* infection, highlighting its potential involvement in disease pathogenesis. While secreted toxins from other bacterial pathogens, such as anthrax and listeriolysin, have previously been detected in infected animals (PMID: 16760326, 7779970), this represents the first such observation in leptospirosis. This finding expands our understanding of leptospiral virulence mechanisms and offers new insight into host–pathogen interactions in this neglected tropical disease. However, the application of VMP detection as a diagnostic tool remains to be validated and was not addressed in this study. We developed a monoclonal antibody-based

immunoassay for circulating VM proteins in blood and urine at 4 days post-infection, a time point that corresponds with the onset of clinical signs in the hamsters experimentally infected with pathogenic *Leptospira*.

14. Line 458. Detecting VM at 4-day post infection in blood is not considered an early detection in the hamster model as the animals are exhibiting morbidity sign at this stage of infection. This should be rephrased.

Lines 494-497

- Thank you for the clarification. We agree that 4 days post-infection corresponds to the onset of morbidity in the hamster model and does not represent an early stage of infection. We have revised the text accordingly to reflect this distinction.
“We developed a monoclonal antibody-based immunoassay for circulating VM proteins in blood and urine at 4 days post-infection, a time point that corresponds with the onset of clinical signs in the hamsters experimentally infected with pathogenic *Leptospira*.”

15. Lines 507-514 and 515-522. The two paragraphs are repetition.

Thank you for pointing this out. We have revised the text to remove the redundancy between the two paragraphs and consolidated the content. **Lines 573-580**

16. Lines 817-818. The arrows in Figure 4B probably correspond to the LA0591 (≈ 48 kDa). Have the other any evidence that the band around 48 kDa correspond to other truncated VMs that could crossreact with the antibody? This should be clarified or corrected.

- Thank you for your observation. The bands at approximately 48 kDa, which likely correspond to LA0591. While we cannot fully exclude the possibility of cross-reactivity with other truncated VMPs of similar size, our antibody was raised specifically against LA0591, and control experiments using recombinant protein support its specificity. We have added a clarification to the result section of this figure:

Lines 431-434

“The ~ 48 kDa band observed in Figure 4B likely corresponds to LA0591. While we cannot fully exclude the possibility of cross-reactivity with other truncated VMPs of similar size, the antibody used was generated against LA0591, and its specificity was supported by control experiments using recombinant protein.

17. Line 853. "2x10⁸ cells/mL" should be corrected to "2x10⁸ leptospire/mL"

Thank you — correction made. **Line 891**

18. Lines 850, 859, "Copenhagni" should be corrected to "Copenhagenii"

- Thank you — correction made at both places. **Line 888**

19. Line 861. "2x10⁸ cells/mL" should be corrected to "2x10⁸ leptospire/mL".

- Thank you — correction made. **Line 898**

20. What is represented in Figure 6C? It is unclear whether the linear regression is made with recombinant LA0591 of Figure 6B? This should be described in the legend.

→ Thank you for your comment. Yes, the linear regression shown in Figure 6C was generated using the recombinant LA0591 protein standard shown in Figure 6B. We have revised the figure legend to clarify this. **Lines 895-897**
“(C) Linear regression was generated using the ELISA data from (B) with recombinant LA0591. The resulting equation is $Y = 1.090 * X + 0.006570$ with $R^2 = 0.9996$.”

21. Line 864. Which patient samples the authors are referring to here?

→ Thank you for pointing this out — correction made, it was a typo error. **Lines 937**

22. Line 866. "VMP proteins" should be corrected into "VM proteins"

Thank you — correction applied. **Lines 908**

23. Lines 875-877. "Clinically, patients affected by leptospirosis often present with a combination of jaundice, renal impairment, and pulmonary complications, all of which stem from the systemic circulation and either primary or secondary action of VMP exotoxins". This statement should be rephrased as there is no experimental evidence that VMs are in fact responsible for all the leptospirosis-associated tissue damage upon acute leptospirosis.

→ Thank you for your thoughtful comment. We agree that, at present, there is no direct experimental evidence definitively linking VMPs to all tissue damage observed in acute leptospirosis. We have revised the statement to reflect this and to avoid overinterpretation of our findings.

Lines 917-920 “Clinically, patients with leptospirosis often present with a combination of jaundice, renal impairment, and pulmonary complications. These manifestations are associated with systemic infection and may involve either primary or secondary action of VMP exotoxins, although their direct role in mediating tissue damage remains to be fully elucidated.”

Re: Spectrum00018-25R2 (**Development of Leptospiral Virulence-Modifying Protein Detection Assay: Implications for Pathogenesis and Diagnostic Test Development**)

Dear Prof. Joseph M. Vinetz:

Your manuscript has been accepted, and I am forwarding it to the ASM production staff for publication. Your paper will first be checked to make sure all elements meet the technical requirements. ASM staff will contact you if anything needs to be revised before copyediting and production can begin. Otherwise, you will be notified when your proofs are ready to be viewed.

Sincerely,
Denis Sereno
Editor
Microbiology Spectrum